# High-throughput laboratory evolution reveals evolutionary constraints in *Escherichia coli*

Tomoya Maeda [1,4✉], Junichiro Iwasawa [2,4], Hazuki Kotani[1], Natsue Sakata[1], Masako Kawada[1], Takaaki Horinouchi[1], Aki Sakai[1], Kumi Tanabe[1] & Chikara Furusawa [1,2,3✉]

Understanding the constraints that shape the evolution of antibiotic resistance is critical for predicting and controlling drug resistance. Despite its importance, however, a systematic investigation of evolutionary constraints is lacking. Here, we perform a high-throughput laboratory evolution of *Escherichia coli* under the addition of 95 antibacterial chemicals and quantified the transcriptome, resistance, and genomic profiles for the evolved strains. Utilizing machine learning techniques, we analyze the phenotype–genotype data and identified low dimensional phenotypic states among the evolved strains. Further analysis reveals the underlying biological processes responsible for these distinct states, leading to the identification of trade-off relationships associated with drug resistance. We also report a decelerated evolution of β-lactam resistance, a phenomenon experienced by certain strains under various stresses resulting in higher acquired resistance to β-lactams compared to strains directly selected by β-lactams. These findings bridge the genotypic, gene expression, and drug resistance gap, while contributing to a better understanding of evolutionary constraints for antibiotic resistance.

[1] RIKEN Center for Biosystems Dynamics Research, 6-2-3 Furuedai, Suita, Osaka 565-0874, Japan. [2] Department of Physics, The University of Tokyo, 7-3-1 Hongo, Tokyo 113-0033, Japan. [3] Universal Biology Institute, The University of Tokyo, 7-3-1 Hongo, Tokyo 113-0033, Japan. [4]These authors contributed equally: Tomoya Maeda, Junichiro Iwasawa. ✉email: tomoya.maeda@riken.jp; chikara.furusawa@riken.jp

The emergence of antibiotic resistance and multidrug-resistant bacteria is a growing global health concern[1–4], and alternative strategies for suppressing the emergence of resistant bacteria are actively being sought. Various mechanisms for antibiotic resistance have been identified, including the activation of efflux pumps, modifications of specific drug targets, and shifts in metabolic activities[5–10]. Quantitative studies of resistance evolution showed that these mechanisms for resistance are tightly interconnected, as demonstrated by the complicated networks of cross-resistance and collateral sensitivity among drugs[11–15], which is the phenomena whereby the acquisition of resistance to a certain drug is accompanied by resistance or sensitivity to another drug. Such interactions among resistance mechanisms result in constraints on accessible phenotypes in evolution[16–18]. For example, the cyclic or simultaneous use of two drugs with collateral sensitivity, to which pathogens did not easily acquire resistance simultaneously, were demonstrated to suppress resistance evolution[19,20]. Thus, elucidating evolutionary constraints are crucial for predicting and controlling the evolution of antibiotic resistance; however, despite its importance, a systematic investigation of evolutionary constraints for antibiotic resistance evolution is still lacking.

Laboratory evolution associated with genotype sequencing and phenotyping is an effective approach to investigate constraints in adaptive evolution[21–23]. Here, we perform high-throughput laboratory evolution of *Escherichia coli* under 95 heterogeneous stressors (Supplementary Data 1). To analyze the expanded cross-resistance/collateral sensitivity network, including both antibiotic and non-antibiotic stressors, while elucidating the molecular mechanisms associated with resistance acquisition, we choose a variety of antibacterial chemicals, including antibiotics with various mechanisms of action, and non-antibiotic toxic chemicals, against *E. coli*. We quantify changes in the transcriptome, genomic sequence, and resistance profile in the evolved strains, producing a multiscale dataset for analyzing stress resistance. By analyzing the gene expression-resistance map through machine-learning techniques, we show the emergence of low dimensional phenotypic states in the evolved strains, indicating the existence of evolutionary constraints. We then analyze the underlying biological processes corresponding to each state by introducing the representative mutations to the parent strain. To examine whether the whole population or only a subset of cells are phenotypically resistant, we conduct a population analysis profile (PAP). Heteroresistance is a common phenomenon for several bacterial species, and antibiotic classes, in which a subpopulation among susceptible cells exhibits increased resistance[24–27]. We identify many heteroresistance-conferring mutations, as well as known repressors for multidrug efflux pumps. We also report decelerated evolution, in which the resistance of the evolved strains in a certain stressor is overtaken by strains evolved in a different stressor. Herein, we demonstrate how our experimental system could provide a quantitative understanding of evolutionary constraints in adaptive evolution, leading to the basis for predicting and controlling antibiotic resistance.

## Results

**Laboratory evolution of *E. coli* under 95 stress conditions**. To systematically investigate drug-resistant phenotypes, we performed high-throughput laboratory evolution using an automated culture system (Fig. 1a)[28] for 95 stressors covering a wide range of action mechanisms (Fig. 1b and Supplementary Data 1). To evaluate the reproducibility of the evolutionary dynamics, six independent culture lines were propagated in parallel for each stressor. In total, 576 independent culture series were maintained (95 stressors plus a control without any stressor × six replicates)

for 27 daily passages corresponding to ~250–280 generations. Figure 1a shows examples of the time course of half-maximal inhibitory concentrations ($IC_{50}$s) during laboratory evolution, while all-time courses of $IC_{50}$s are shown in Supplementary Fig. 1. Among the 95 stressors, a significant increase in $IC_{50}$ was observed for 89 stressors (Mann–Whitney $U$ test, false discovery rate (FDR) < 5%). For further phenotypic and genotypic analyses, we selected 192 evolved strains, i.e., four evolved strains isolated from 47 stressors plus a control without any stress. These 47 stressors were selected from the initial 95 stress environments, due to the limitation of experimental capacity. Selections were made based on the degree of increased $IC_{50}$ values; to ensure a variety of stressors with different action mechanisms, and based on the predicted novelty of the expected results. For further analysis, we selected the top four independent culture lines showing higher $IC_{50}$ values among the six.

**Phenotypic and genotypic changes in evolved strains**. To explore phenotypic changes in the 192 evolved strains, we first quantified changes in the stress resistance profiles by measuring the $IC_{50}$ of all 47 chemicals for each evolved strain (9024 measurements in total), relative to the parent strain (Supplementary Data 2 and "Methods"). The resistance profile measurements allowed us to study how common cross-resistance, a phenomenon where an evolved strain in certain stress gains resistance to another stress, occurs. Here, cross-resistance refers to both cross-genetic resistance and cross-heteroresistance. By comparing the four evolved replicates and the parent strain, we found that 336 and 157 pairs of stressors exhibited cross-resistance and collateral sensitivity, respectively, within the possible 2162 combinations (Mann–Whitney $U$ test, FDR < 5%, Supplementary Fig. 2).

To investigate genetic alterations underlying the observed resistance, we performed genome resequencing analysis of the 192 evolved strains (Supplementary Data 3). Although some of these strains carried more than 20 mutations, 147/192 evolved strains (76.6%) harbored fewer than five. To show the variation in the number of mutations among strains evolved in the same environment, the mean number and standard deviation of mutations for the four strains are shown in Supplementary Fig. 3. Among the 47 stressors, the highest number of mutations was observed in glutamic acid γ-hydrazide (GAH) evolved strains carrying $157 \pm 67$ mutations (Supplementary Fig. 3), which was significantly more than the number observed in evolved strains against known mutagens (e.g., 4-nitroquinoline-1-oxide (NQO, $23 \pm 5$ mutations) and mitomycin C (MMC, $27 \pm 4$ mutations)), indicating a high mutagenic activity of GAH. Excluding strains evolved in GAH, NQO, and MMC, which harbored more than 18 mutations on average per stressor, 21 and 307 mutations were identified as synonymous and nonsynonymous mutations, respectively (Fig. 1c). For strains evolved in stresses other than GAH, NQO, and MMC, the ratio of nonsynonymous to synonymous mutations per site was 5.26, implying that approximately 80% of the nonsynonymous mutations were beneficial[29,30].

**Supervised principal component analysis (PCA) reveals modular phenotypic states**. Phenotypic changes of the evolved strains were quantified by transcriptome analysis to examine gene expression levels responsible for stress resistance (Supplementary Data 4). In the transcriptome analysis, all evolved strains were cultured without the addition of stressors to standardize the culture condition. To explore the relationship between gene expression changes and resistance evolution, we performed dimension reduction on the gene expression data using supervised PCA, which enables the extraction of a subspace in which

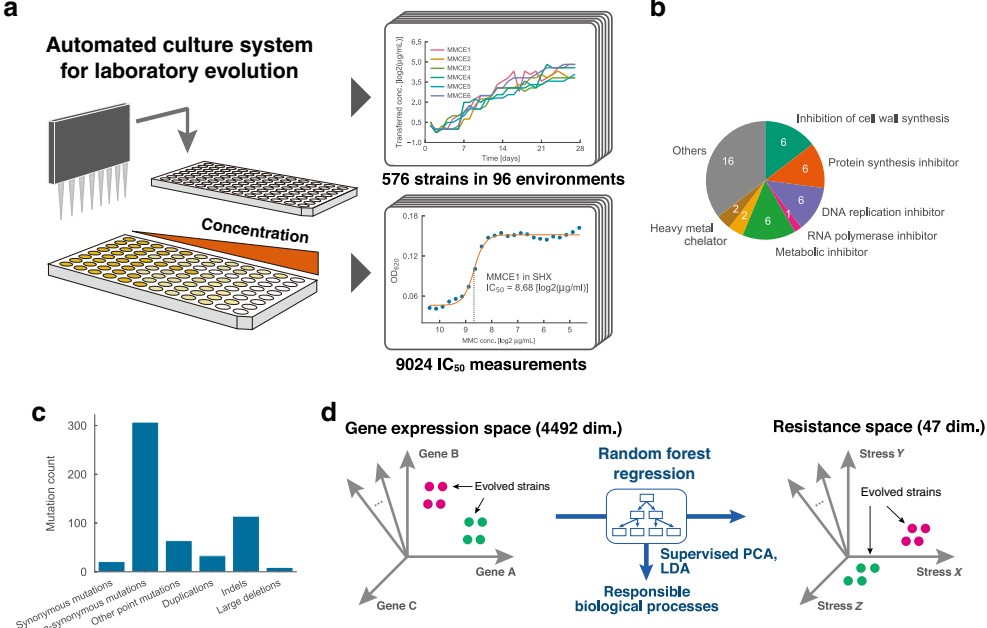

**Fig. 1 Laboratory evolution of *E. coli* under 95 stress conditions. a** Schematic of the experimental setup (automated culture system for laboratory evolution). **b** Stress categories for the environments used in the half-maximal inhibitory concentration ($IC_{50}$) measurements. **c** Distribution of mutation events for the evolved strains according to its mutation type, except for strains evolved in glutamic acid γ-hydrazide (GAH), 4-nitroquinoline-1-oxide (NQO), and mitomycin C (MMC). Other point mutations include those in intergenic/noncoding regions. Source data are provided as a Source Data file. **d** A random forest regression model was constructed to predict $IC_{50}$ values using the gene expression levels ("Methods"). Supervised principal component analysis (PCA) was applied to the 213 gene expression levels selected by the random forest algorithm.

the dependency between gene expression and stress resistance is maximized[31]. First, to exclude genes with unchanged or noisy expression in the evolved strains, we used random forest regression to screen the genes that contribute to the prediction of resistance changes (Fig. 1d), which resulted in the selection of 213 genes with high correlations to resistance levels. Supervised PCA based on the expressions of these genes revealed the existence of clusters of evolved strains in the dimension reduced gene expression space (Supplementary Fig. 4). To clarify the clusters, we applied hierarchical clustering to the evolved strains (Fig. 2a and Supplementary Fig. 5), which demonstrated modular classes of expression profiles. Intriguingly, strains within the same class were not necessarily evolved in the same stress, nor stress category. Similar phenotypic convergence of drug-resistant strains has recently been reported for *Pseudomonas aeruginosa*[18]. To elucidate characteristic gene expression for each class, we applied linear discriminant analysis (LDA), which allowed us to extract the most discriminative set of genes for each class, through the observation of each decision boundary (Fig. 2b).

To investigate how the classes of gene expression profiles correspond to stress resistance, we observed the relative $IC_{50}$ for each of the 47 stresses of each evolved strain sorted based on the hierarchical clustering of gene expressions (Fig. 2c). As shown in the figure, the classes in the supervised PCA space correspond well with the stress resistance patterns. To evaluate how accurate the neighboring relationship in the resistance space corresponds to that of the supervised PCA space, we computed the class dissimilarity ($W_n$) in the resistance profile for the classes in the dimension reduced gene expression space. For comparison, we computed $W_n$ for classes based on the resistance space, the whole gene expression space, and the genotype space (Supplementary Fig. 6). This metric revealed that clustering in the supervised PCA space corresponds well with the resistance space, and better than that of the whole expression space and genotype space. This indicates that topological relationships between the strains in the

resistance space could be accurately represented in a subspace of the gene expression space. In contrast, direct clustering in the resistance space did not necessarily correspond with characteristic gene expression profiles. When we calculated the class dissimilarity $W_n$ in the gene expression space based on the clustering result in resistance space, the value approached that of random clustering (Supplementary Fig. 6b). This was likely due to the effective degrees of freedom in the resistance space not being sufficient to recover necessary information in the gene expression space, while the supervised PCA space obtained good representations conserving information for both the expression and resistance space (Supplementary Fig. 6).

**Relationship between mutations and supervised PCA classes**. We next sought to determine how the identified classes in the supervised PCA classes could be characterized. To our surprise, relatively clear relationships between the genome, transcriptome, and resistance profiles were observed for the modular classes in the supervised PCA space. Figure 2d shows a subset of the commonly mutated genes within the 192 evolved strains. As shown, the patterns of fixed mutations coincide well with the modular classes in the gene expression space, although no genotypic information was used for the hierarchical clustering. This suggests that the identified mutations play a meaningful role in the modular gene expression classes. For example, all evolved strains in class 1 had mutations in *mprA*, which encodes a repressor for multidrug-resistance pump EmrAB, while all strains in class 11 had mutations in *prlF*, which encodes the antitoxin for the PrlF (SohA)-YhaV toxin–antitoxin (TA) system.

Although strains in the same class had similar gene expression levels, they did not necessarily share the same mutations. For example, evolved strains in class 5 showed an increased expression of *acrB*, which encodes a component of the AcrAB/TolC multidrug efflux pump, while most of the strains (26/28) in class 5 had mutations in *acrR*, which encodes a repressor for

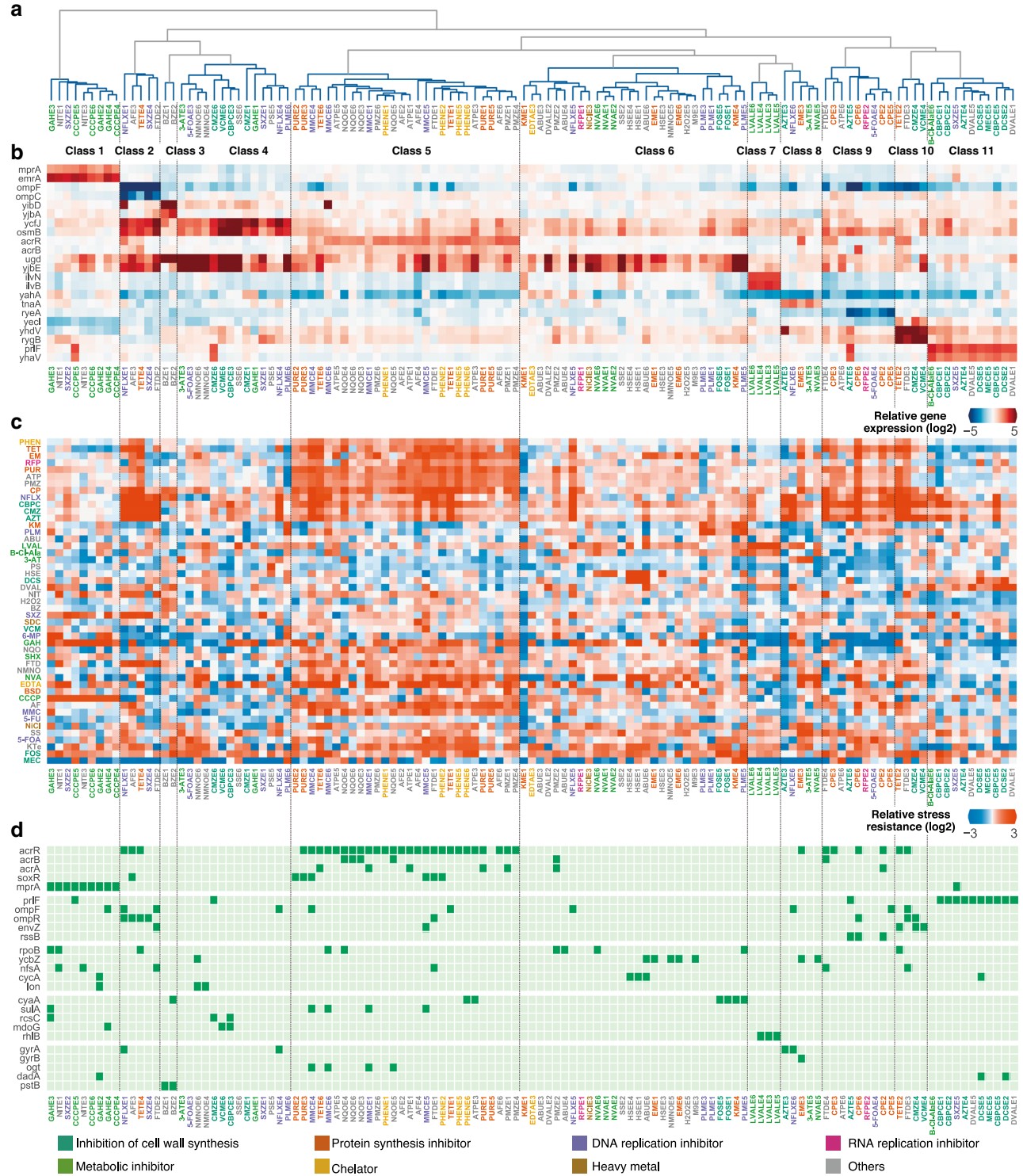

**Fig. 2 Supervised principal component analysis (PCA) reveals distinct clusters in the genotype, expression, and the resistance (IC$_{50}$) space. a** Dendrogram of the result of hierarchical clustering performed in the 36-dimensional supervised PCA space. One cluster and three singletons were omitted due to visibility. The full version is presented in Supplementary Fig. 5. **b** Gene expression levels of representative genes for each cluster, relative to the parent strain. The genes were selected from the intersection of the top two gene weights for the linear discriminant analysis (LDA) axis and differentially expressed genes ("Methods"). **c** IC$_{50}$ values relative to the parent strain. Colors for the tick labels correspond to the stress categories. **d** Characteristic mutated genes for each class. Mutated genes enriched for each cluster clarified by Fisher's exact test ($P < 0.01$) are presented. Mutated genes that were identified in more than seven strains are also presented. Genes are sorted based on gene ontology categories. Source data are provided as a Source Data file.

*acrAB* (Fig. 2b, d). Interestingly, the other two strains also showed an increase in *acrB* expression without an *acrR* mutation. Meanwhile, strains in class 8 consistently had increased expression of *tnaA*-encoding tryptophanase, whereas four out of five strains had mutations in genes encoding DNA gyrase subunit A or B (*gyrA* or *gyrB*, Fig. 2b, d). We further confirmed, through quantitative reverse transcription-polymerase chain reaction (qRT-PCR) analysis, that the introduction of the observed H45Y mutation in *gyrA* to the parent strain leads to an 11.1 ± 6.6-fold increase in *tnaA* mRNA level. Consistent with these results, a previous study suggested the involvement of a mutation in *gyrB* for increased TnaA production and quinolone resistance[32]. Note, NVAE5, which did not have a mutation in *gyrA* nor *gyrB*, also showed an increased expression of *tnaA*. These results suggest the existence of multiple paths in the genotypic space for *E. coli* to reach desired expression and resistance levels.

Commonly decreased expression of *ompF*, which encodes the outer membrane porin, was observed in several classes (class 2, 9, and 10, Fig. 2d) and was associated with resistance to cell wall inhibitors and other stresses. A decrease in *ompF* expression can be caused by either inactivation of the OmpR/EnvZ two-component system or RssB, which is a regulator of the alternative sigma factor RpoS[33–35]. Indeed, all strains in class 2 and class 10 had mutations in either *ompR* or *envZ*, and four out of nine strains in class 9 had mutations in *rssB*. Although these strains commonly had a decrease in the *ompF* expression level, they have been assigned to different classes based on their consistently different gene expression profiles for genes such as the *ompC*, encoding a porin, and *rygB*, encoding a small RNA involved in the regulation of the outer membrane composition (Fig. 2b). Interestingly, although all three classes showed resistance to β-lactams (e.g., carbenicillin (CBPC), and cefmetazole (CMZ)), resistance levels to stresses such as sulfisoxazole (SXZ) and DL-3-hydroxynorvaline (NVA) differed between classes (e.g., strains in class 2 and 10 exhibited resistance to SXZ, while strains in class 9 did not, Fig. 2c, Supplementary Fig. 2). This suggests that different classes in the supervised PCA space correspond to different stress resistance mechanisms.

Moreover, evolved strains derived from the same selection pressure were not always categorized in the same class. For example, none of the four SXZ evolved strains shared the same supervised PCA class (SXZE2 in class 1, SXZE4 in class 2, SXZE1 in class 4, and SXZE5 in class 11), and each strain showed different expression and resistance patterns, indicating a rugged fitness landscape with multiple local peaks. This ruggedness was observed for the four norfloxacin (NFLX) evolved strains as well. Intriguingly, these local peaks were accessible not only by SXZ and NFLX evolved strains but also by strains that evolved in other stresses (for example, see class 2). These results suggest that evolution under the same selection pressure does not necessarily lead to the same phenotype[36,37] and that these local peaks in the fitness landscape are shared by different stresses. Overall, the low dimensional phenotypic states revealed by our high-throughput measurements loosely corresponded with the genotypic space, suggesting the existence of various genotypic pathways to reach local optima in the fitness landscape, which were shared by strains evolved in diverse stresses.

**Commonly mutated genes provide the basis for chemical resistance**. Genome resequencing analysis revealed that several genes were commonly mutated in multiple evolved strains, suggesting the contribution of these mutations to the observed resistance acquisition. Commonly mutated genes were defined as mutations in the same gene identified in a minimum of two of the

four independent culture lines evolved under the same environment. The detailed description of these genes and mutations are presented in Supplementary Discussion. To verify the effects of the commonly mutated genes found in the evolved strains, we introduced 64 of the representative mutations (Supplementary Data 2) to the parent strain by multiplex automated genome engineering (MAGE)[38], and quantified changes in the IC$_{50}$s of all 47 chemicals against each strain. We first asked whether the cross-resistance and collateral sensitivities observed within the evolved strains could be reproduced by the 64 reconstructed mutant strains. Accordingly, we calculated Pearson's correlation coefficient $R$ between the IC$_{50}$s of all 47 stresses within the evolved strains. We recognized that some stress pairs showed a high correlation in their resistance of the evolved strains. For example, evolved strains resistant to CBPC tended to exhibit resistance to aztreonam (AZT) as well ($R = 0.95$, Fig. 3a), both of which constitute β-lactam stresses. We then calculated correlation coefficients for the reconstructed mutant strains. Certain stress pairs, such as TET and B-Cl-Ala showed a negative correlation not only for the evolved strains but also for the reconstructed mutant strains (Fig. 3b). We then compared the coefficients of the mutant strains with that of the evolved strains. The coefficients for the evolved strains highly correlated with those of the reconstructed mutant strains ($R = 0.66$, Fig. 3c, d), indicating that the observed collateral relationships within the evolved strains could be explained by the commonly mutated genes. Interestingly, a high correlation was observed between the evolved strains and mutant strains excluding transporter related mutations (e.g., *acrR*, *ompF*), indicating that the high correlation in phenotype is not only caused by transport, which is a major mechanism of resistance (Supplementary Fig. 7). These results allowed us to further investigate the role of mutations in the context of cross-resistance and collateral sensitivities.

To examine whether the commonly identified mutations confer genetic resistance or heteroresistance, we conducted a PAP of 33 drug combinations and the reconstructed mutant strain pairs (Supplementary Fig. 8). These combinations were selected according to the mutation and drug pairs exhibiting >eightfold increase in IC$_{50}$ value, the representative mutations in the supervised PCA classes (*acrR* in class 5, *gyrA* in class 8, *mprA* in class 1, *ompF* in class 2 and 10, *prlF* in class 11, and *rssB* in class 9), and the resistance-conferring mutations that have not been reported yet (*gshA*, *ycbZ* and *yhjE*). Heteroresistance strains are defined as strains with resistant subpopulations growing at frequencies of $1 \times 10^{-7}$ or higher in antibiotic concentrations at least 8-fold higher than the highest non-inhibitory concentration of the wild-type[27,39]. Among the 33 combinations, 11 were categorized as genetic resistance, including seven cross-genetic resistances; while 15 combinations were categorized as hetero-resistance, including 12 cross-heteroresistance pairs (Supplementary Fig. 8). Although seven combinations exhibited a twofold increase compared to the highest non-inhibitory concentration, there were no resistant subpopulations at antibiotic concentrations eightfold higher. Regarding the supervised PCA classes, the *ompF* mutant showed cross-genetic resistance, while the *acrR*, *mprA*, *prlF*, and *rssB* mutant strains exhibited cross-heteroresistance (Supplementary Fig. 8). Alternatively, the *gyrA* mutant strains showed both cross-genetic resistance and cross-heteroresistance (Supplementary Fig. 8). These results suggest that heteroresistance is a common resistance phenotype among our laboratory evolved strains, which agrees with the results of previous studies using clinical isolates[26,27].

Most of the evolved strains acquired cross-resistance from mutations in genes encoding transporters and porins (Table 1 and Supplementary Data 5). Antibiotic resistance of *E. coli* can be triggered by the overexpression of efflux systems and decreased

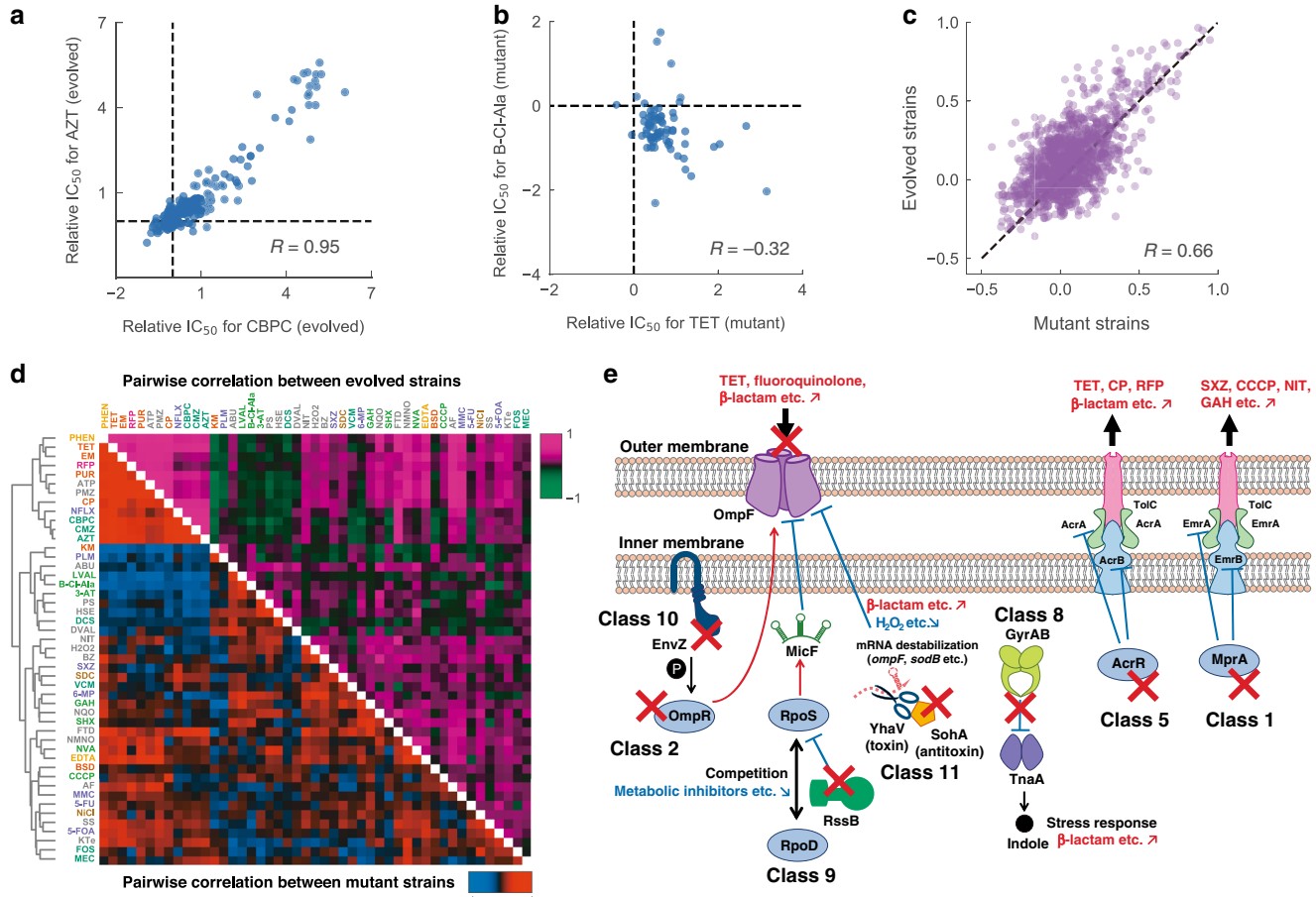

**Fig. 3 Commonly mutated genes provide the basis for chemical resistance. a, b** Relationship between the $IC_{50}$ values of carbenicillin/aztreonam (CBPC/AZT) and tetracycline/β-chloro-L-alanine (TET/B-Cl-Ala) for the 192 evolved strains and 64 site-directed mutants, respectively. *R* denotes the Pearson's correlation coefficient. **c** Relationship between the corresponding pairwise correlation coefficients shown in panel **d**. **d** Pearson's correlation coefficient for all pairwise combinations of stress resistance for the evolved strains (upper right) and the site-directed mutants (lower left). The order of stresses was determined by hierarchical clustering performed on the pairwise correlation values of the site-directed mutants. **e** Schematic illustration of stress resistance acquisition mechanisms corresponding to the supervised principal component analysis (PCA) clusters. Typical stresses which exhibited resistance (red) and sensitivity (blue) are shown. Source data are provided as a Source Data file.

production of porin proteins[6,34,39]. In this study, by expanding the quantification of cross-resistance/collateral sensitivity network to various stressors including antibiotics and non-antibiotics, we confirmed the contribution of altered uptake/efflux activities in cross-resistance relationships among them (Fig. 2d, Table 1, and Supplementary Data 5). For example, activation of the multidrug efflux pump AcrAB/TolC, through the inactivation of its repressor AcrR, resulted in resistance not only to previously described substrates such as tetracycline (TET) and erythromycin (EM)[6,40,41], which are both known as protein synthesis inhibitors but also to substrates that have not been reported yet such as NVA (threonine analog) and NQO (mutagen) (for a detailed list see Table 1). The activation of EmrAB/TolC is regulated by *mprA*, which is also an efflux pump, and results in resistance to a previously identified substrate CCCP (uncoupling agent)[42], as well as to substrates that have not been reported yet, such as chloramphenicol (CP, protein synthesis inhibitor) and phleomycin (PLM, DNA intercalator) (Fig. 3e and Supplementary Data 5). Since the reconstructed *acrR*, and the *mprA* mutant strains, exhibited cross-heteroresistance to EM and TET, or GAH and TET respectively (Supplementary Fig. 8), the enhanced drug efflux caused by *acrR* or *mprA* mutations can cause cross-heteroresistance. Similarly, it was previously reported that overexpression of the efflux pump results in heteroresistance

in several pathogenic bacteria[24]. We also confirmed that inactivation of the OmpF porin resulted in resistance to previously described substrates such as CBPC (cell wall synthesis inhibitor)[5,43], and substrates that have not been reported yet such as 1,10-phenanthroline (PHEN, chelator), puromycin (PUR, protein synthesis inhibitor), and other chemicals (Fig. 3e and Table 1). Our results suggest that mutations in the transporters and porins above could lead to resistance to a wide spectrum of drugs with different mechanisms of action. Since the reconstructed *ompF* mutant strain showed cross-genetic resistance to CBPC, FTD, and NVA (Supplementary Fig. 8), inactivation of the OmpF porin can cause cross-genetic resistance. We also identified the contribution of an uncharacterized transporter to chemical resistance. Two D-valine (DVAL) evolved strains contained mutations in *yhjE*, which encodes a putative transporter. The contribution to resistance was confirmed through the reconstructed *yhjE* inactivation mutant strain which showed a more than twofold increased resistance to PLM, NVA, and DVAL (Supplementary Data 2), suggesting uptake through YhjE. Moreover, the DVAL and NVA resistances by *yhjE* inactivation constituted heteroresistance (Supplementary Fig. 8). Taken together, our results indicate that the effects of chemical uptake and efflux are major mechanisms for cross-genetic resistance and heteroresistance.

**Table 1 Representative cross-resistances and collateral sensitivities observed in the reconstructed strains.**

| Mutation | Resistance | Sensitivity |
|---|---|---|
| *acrR* | Chloramphenicol, rifampicin, **cefmetazole**, **aztreonam**, acriflavine, **nickel (II) chloride**, carbenicillin, mitomycin C, **amitriptyline**, **1,10-phenanthroline**, DL-**3-hydroxynorvaline**, tetracycline, **4-nitroquinoline 1-oxide**, **promethazine**, **nitrofurantoin**, **sodium salicylate**, **furaltadone**, erythromycin, puromycin, **5-fluoroorotic acid**, *N*-methyl-*N*-octylamine | D-cycloserine, L-homoserine, D-valine |
| *mprA* | **Chloramphenicol**, **rifampicin**, **5-fluorouracil**, aztreonam, **6-mercaptopurine**, **mitomycin C**, **phleomycin**, DL-**3-hydroxynorvaline**, **tetracycline**, DL-**serine hydroxamate**, **promethazine**, **blasticidin S**, **L-glutamic acid γ-hydrazide**, **erythromycin**, CCCP, DL-**2-aminobutyric acid**, **puromycin**, **5-fluoroorotic acid**, *N*-methyl-*N*-octylamine | |
| *ompF* | Chloramphenicol, rifampicin, cefmetazole, aztreonam, **nickel (II) chloride**, carbenicillin, norfloxacin, **phleomycin**, **1,10-phenanthroline**, DL-**3-hydroxynorvaline**, mecillinam, tetracycline, **sodium salicylate**, **blasticidin S**, **furaltadone**, erythromycin, **puromycin**, *N*-methyl-*N*-octylamine | D-cycloserine, D-valine |
| *prlF* | Chloramphenicol, aztreonam, kanamycin, carbenicillin, sodium dichromate, D-cycloserine, phleomycin, mecillinam, puromycin, D-valine, 5-fluoroorotic acid | Rifampicin, hydrogen peroxide, nickel(II) chloride, benserazide, 4-nitroquinoline 1-oxide, DL-serine hydroxamate, L-glutamic acid γ-hydrazide |
| *rssB* | Chloramphenicol, rifampicin, cefmetazole, aztreonam, acriflavine, carbenicillin, sodium dichromate, amitriptyline, phleomycin, tetracycline, promethazine, sodium salicylate, blasticidin S, erythromycin, puromycin, *N*-methyl-*N*-octylamine | L-valine, β-chloro- L-alanine, nickel(II) chloride, protamine sulfate, D-cycloserine, 3-amino-1,2,4- triazole, 4-nitroquinoline 1-oxide, DL-serine hydroxamate, L-glutamic acid γ-hydrazide, L-homoserine |
| *ycbZ* | Chloramphenicol, aztreonam, 6-mercaptopurine, nickel (II) chloride, carbenicillin, amitriptyline, phleomycin, DL-3-hydroxynorvaline, tetracycline, EDTA, promethazine, sodium salicylate, blasticidin S, erythromycin, puromycin, 5-fluoroorotic acid, *N*-methyl-*N*-octylamine | D-cycloserine, 4-nitroquinoline 1-oxide |
| *yhjE* | **Acriflavine**, **carbenicillin**, **norfloxacin**, **phleomycin**, DL-**3-hydroxynorvaline**, **puromycin**, **D-valine** | Nickel (II) chloride, DL-serine hydroxamate |

Chemicals that were identified as significantly increased or decreased IC$_{50}$ values (Mann–Whitney *U* test, FDR < 5%) in the reconstructed *acrR*, *mprA*, *ompF*, *prlF*, *rssB*, *ycbZ*, and *yhjE* mutant strains are shown, respectively. For transport machinery (i.e., *acrR*, *ompF*, *mprA*, and *yhjE*), newly identified putative substrates are shown in bold letters. The full list of cross-resistances and collateral sensitivities observed in all 64 reconstructed mutant strains are shown in Supplementary Data 5.

We have also identified mechanisms underlying cross-resistance to chemicals. The increased expression of PrlF-YhaV TA system along with mutations in *prlF* was observed in class 11 (Fig. 2b, d). Interestingly, all evolved strains carrying the *prlF* mutation, including the 11 strains in class 11, had the same mutation: i.e., duplication of TTCAACA sequences located 272 bp downstream of the start codon (Supplementary Data 3). Although the contribution of PrlF-YhaV to stress resistance has not yet been reported, these evolved strains, and the reconstructed *prlF* mutant strain, commonly exhibited resistance to CBPC, AZT, and DVAL (Fig. 2c and Table 1). Furthermore, the *prlF* mutant strain showed cross-heteroresistance to 5-FOA and CBPC indicating that the *prlF* mutation conferred heteroresistance (Supplementary Fig. 8). All 11 strains in class 11 showed a decreased expression of *ompF* (Fig. 2b), and the decrease in *ompF* mRNA level by 0.52 ± 0.01 in the reconstructed *prlF* mutant strain was confirmed by qRT-PCR analysis. These results indicate that cross-resistance to CBPC, AZT, and DVAL by the *prlF* mutation is at least partially caused by decreased expression of *ompF*. Since YhaV is a translation-dependent RNase[44], this decrease in *ompF* expression might be caused by the alteration of global gene expression.

The contribution of the uncharacterized protein YcbZ to cross-resistance was also confirmed. YcbZ is a putative protease shown to be involved in translation and ribosome biogenesis[45]. We identified YcbZ mutations in three EM evolved strains and two *N*-methyl-*N*-octylamine (NMNO) evolved strains. These strains, as well as the reconstructed *ycbZ* inactivation mutant strain, commonly exhibited resistance not only to EM and NMNO, but also to ethylenediamine-*N,N,N',N'*-tetraacetic acid, disodium salt, dihydrate (EDTA), NVA, and 5-fluoroorotic acid monohydrate (5-FOA; Fig. 2c and Table 1). Moreover, the *ycbZ* mutant strain showed cross-heteroresistance to EDTA and NVA, indicating that the *ycbZ* mutation confers heteroresistance (Supplementary Fig. 8). Note, one out of four control experiments without any additional stressor (M9E3 strain) also carried the *ycbZ* mutation for EM. Additionally, ABUE6 evolved strains carried the *ycbZ*

mutation although the *ycbZ* disruption did not confer ABU resistance (Supplementary Table S2). These mutations in M9E3 and ABUE6 strains may have arisen from the application of 1/10 IC$_{50}$ erythromycin to all culture medium, to avoid contamination, which may have caused adaptive evolution to occur in response to the low concentrations of erythromycin.

**Quantification of collateral sensitivities.** The identified classes in the supervised PCA space revealed collateral sensitivity relationships for antibiotic resistance acquisition. For example, the evolved strains in class 9 exhibited sensitivity to metabolic inhibitors such as L-valine (LVAL), β-chloro-L-alanine (B-Cl-Ala), 6-mercaptopurine monohydrate (6-MP), GAH, and 3-amino-1,2,4-triazole (3-AT) (Fig. 2c and Supplementary Fig. 2). Since 4/9 of the class 9 strains had mutations in *rssB*, we speculated that this observed sensitivity is caused by *rssB*. This was confirmed through IC$_{50}$ measurements of the reconstructed *rssB* inactivation mutant strain which showed a two- to fivefold change in sensitivity to the stresses above (Supplementary Data 2 and Supplementary Fig. 9). Moreover, since class 9 strains and the reconstructed *rssB* strain both show resistance to cell wall inhibitors and other stresses (AZT, CBPC, TET), our results indicate a trade-off between these stresses and metabolic inhibitors. It has been reported that while *E. coli* strains with higher *rpoS* levels show increased resistance to several external stresses[46], they also exhibit decreased carbon source availabilities and poor competitiveness for low concentrations of nutrients due to the competition between RpoS and the house-keeping sigma factor RpoD (sigma 70)[47]. Since RssB facilitates the degradation of RpoS, the collateral sensitivities to several metabolic inhibitors in class 9 evolved strains could also be caused by the sigma factor competition.

Collateral sensitivities associated with the *prlF* mediated resistance were also identified. All evolved strains in class 11, which carry the same *prlF* mutation, exhibited collateral sensitivity to hydrogen peroxide (H$_2$O$_2$), benserazide (BZ), and NQO (Fig. 2c and Supplementary Fig. 2). Consequently, the same

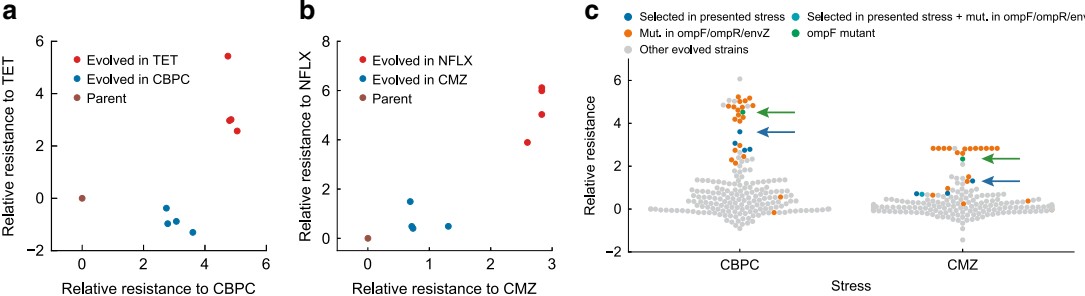

**Fig. 4 Decelerated evolution against β-lactam antibiotics. a, b** Decelerated evolution observed within the evolved strains. Relative $\log_2$ ($IC_{50}$) for evolved strains in carbenicillin (CBPC) and tetracycline (TET) (**a**), and relative $\log_2$ ($IC_{50}$) for evolved strains in norfloxacin (NFLX) and cefmetazole (CMZ) (**b**). **c** Relative $IC_{50}$ values for CBPC and CMZ for all 192 evolved strains. Many of the strains which exhibit resistance higher than the CBPC and CMZ evolved strains had a mutation in *ompF* or its regulators *ompR* and *envZ* (orange). The CBPC and CMZ resistance of the *ompF* introduced strain also exhibited higher resistance (green, green arrow) than the CBPC and CMZ evolved strains (blue, cyan, denoted by a blue arrow). Source data are provided as a Source Data file.

collateral sensitivities were also observed in the reconstructed *prlF* mutant strain (Supplementary Fig. 9). These results suggest that DVAL, CBPC resistance, acquired through the *prlF* mutation, leads to a trade-off for $H_2O_2$, BZ, and NQO. Previous studies reported that *E. coli* mutant strains lacking superoxide dismutase showed increased susceptibility to $H_2O_2$ mediated killing[48]. Indeed, all strains in class 11 consistently exhibited a 0.45 ± 0.11-fold decrease in *sodB* expression, which encodes (Fe) superoxide dismutase, in the transcriptome data. Through qRT-PCR analysis, the decrease in *sodB* mRNA level by 0.46 ± 0.06 in the reconstructed *prlF* mutant strain was also confirmed. This suggests that the observed $H_2O_2$ sensitivity is caused by the degradation of *sodB* through the YhaV toxin.

**Decelerated evolution against β-lactam antibiotics.** In the adaptive evolution to β-lactams (i.e., CMZ and CBPC), we found that certain strains, evolved under specific stresses, acquired higher resistances to β-lactams than strains that were directly selected by β-lactams. Figure 4 presents $IC_{50}$ values of CBPC and CMZ observed in the 192 evolved strains, showing that evolved strains under CBPC and CMZ (marked by blue and cyan) did not exhibit the highest resistance levels to β-lactams, but rather, evolved strains under other stresses, such as TET and NFLX, showed much higher resistance levels (Fig. 4a, b). This "decelerated" resistance evolution to β-lactams was reflected in the difference in the mutation profile among the evolved strains. We found that the evolved strains that exhibited the highest resistances generally had mutations in genes related to the membrane porin protein OmpF, i.e., *ompF*, *ompR* and *envZ* (Fig. 4c). The disruption of OmpF reportedly contributes to resistance acquisition to β-lactams[5], and in fact, we confirmed that the introduction of the identified *ompF* mutations to the parent strains significantly increased $IC_{50}$ values of CMZ and CBPC (green marker in Fig. 4c). In contrast, the strains evolved under CBPC or CMZ had fewer mutations in OmpF related genes (one out of eight evolved strains) in comparison with other strains with high β-lactam resistance ($P = 0.04$, Fisher's exact test, $N = 32$). This result might suggest that in our laboratory evolution setup, the fixation of mutations related to OmpF is suppressed under the addition of β-lactams, even though they can increase their resistance to the drug. Possible explanations for this decelerated evolution against β-lactam could be the existence of a fitness cost associated with *ompF* mutation, and/or negative epistasis between *ompF* and *prlF* mutations. However, neither such fitness cost nor negative epistasis was observed ("Methods" and Supplementary Fig. 10). Since a link between mutation supply rate and adaptation rate was reported[49], it is also possible that decreased

evolution is caused by the difference in mutation frequencies between β-lactams and other stresses. To address this possibility, we measured mutation frequencies with β-lactams (CBPC and CMZ), or other drugs showing higher β-lactam resistance (CP, NFLX, TET); however, differences were not observed in mutation frequencies (Supplementary Fig. 10). Therefore, at present, the mechanism for the decelerated evolution against β-lactams remains unclear.

**Discussion**

In this study, we performed laboratory evolution of *E. coli* under various heterogenous stress conditions which allowed us to elucidate the molecular mechanisms associated with resistance acquisition to antibiotics and non-antibiotics stressors. Combined with supervised PCA and hierarchical clustering, our high-throughput phenotypic analysis led to the identification of modular phenotypic classes both in the gene expression space and the stress resistance space, suggesting close interactions between changes in gene expression and stress resistance. Furthermore, these classes included strains that evolved in a different type of stresses, indicating the existence of evolutionary constraints that do not necessarily depend on the stressor's mechanism of action.

Our results include valuable information on evolutionary constraints for antibiotic resistance, and thus, provide important insights for alternative clinical strategies. For instance, we found that various antibiotics with different action mechanisms exhibited collateral sensitivity to metabolic inhibitors including L-valine (LVAL), β-chloro-L-alanine (B-Cl-Ala), and glutamic acid γ-hydrazide (GAH). We also found that strains that evolved in various stresses showed collateral sensitivity to $H_2O_2$ and protamine sulfate (PS) (15/47 and 14/47 stresses, respectively). Collateral sensitivity to $H_2O_2$ was commonly observed for strains in class 9 and 11 (Supplementary Fig. 2), which might have been caused by the alteration of global gene expression by mutations in *rssB* (class 9) and *prlF* (class 11) since $H_2O_2$ has many cellular targets[50]. Meanwhile, collateral sensitivity to PS was commonly observed for strains in class 2, 5, and 11 (Supplementary Fig. 2), where enhanced drug efflux caused by *acrR* mutations (class 5), or decrease drug uptake by *ompF* associated mutations (class 2 and 9), may have led to collateral sensitivity to PS, which has been shown to target the cytoplasmic membrane in *Salmonella typhimurium*[51]. These results suggest that the perturbation of metabolic activity, reactive oxygen species generation, and alteration of cytoplasmic membrane permeabilization could serve as possible strategies to suppress antibiotic resistance.

Based on the resequencing analysis of the evolved strains, we clarified the effect of single mutations for resistance acquisition

through the introduction of representative mutations to the parent genome. The pairwise correlation coefficients between stresses, indicating cross-resistance and collateral sensitivity, observed in the reconstructed mutants agreed with those of the evolved strains (Fig. 3c, d). Although correlation coefficients are only capable of probing the averaged directionality evolution, these results suggest that the observed evolutionary constraints for resistance in the evolved strains were rooted in acquired mutations.

The analysis of reconstructed mutant strains also provided valuable information on the genetic basis of resistance acquisition. These results demonstrated that the pattern of resistance acquisition observed in the evolved strains could be characterized by known resistance-conferring genes, such as *acrAB* and *ompF*. Furthermore, the analysis identified genetic mechanisms for the stress resistance that have not been reported yet, such as CBPC, AZT, and DVAL resistance through *prlF* mutation (Fig. 3e), which is related to the TA system. We also discovered a contribution made by uncharacterized genes to resistance acquisition, such as the *ycbZ* and *yhjE* mutations causing EM, NMNO, EDTA, NVA, and 5-FOA resistances, and PLM, NVA, and DVAL resistances, respectively. Although we only highlighted a limited number of resistance mechanisms, a comprehensive description of the mutations identified in the stresses used in this study is given in the Supplementary Discussion. We believe that sharing our results in this manuscript, including identified mutations, transcriptome changes, and resistance profiles in the evolved strains, as well as phenotypic changes in the reconstructed mutants, will provide clues for future studies and contribute to the field of antibiotic resistance evolution. Of course, these findings can be affected by various conditions, including the strength of selection pressure, mutation frequency, and culture condition. The problem of how the identified genotypic and phenotypic alterations are affected by other conditions will be an important topic and remain as future works.

Note, the analyses presented in this study are not without limitations. First, the identification of distinct classes in phenotypic changes (Fig. 2) was based on gene expression changes, and thus, the analysis did not detect resistance acquiring mechanisms with small gene expression changes. For example, in the evolved strains under 6-MP, mutations in the *hpt* gene, encoding hypoxanthine phosphoribosyltransferase, were commonly fixed (three out of four evolved strains), suggesting a contribution of the mutation to the 6-MP resistance phenotype. However, these 6-MP resistant strains exhibited little expression changes compared to the parent, and thus, we could not identify the common phenotypic changes through gene expression-based analysis. Such evolved strains with minor expression changes were assigned to class 12 (Supplementary Fig. 5). Including mutations fixed in these strains, a detailed description of the mutations conferring resistance for each stress is given in Supplementary Data 3. Second, the introduction of single mutations fixed in the evolved strains was not always sufficient to explain the resistance changes observed in the laboratory evolution. For example, all evolved strains in class 1 (Fig. 2) had mutations in *mprA*, strongly suggesting the contribution of this mutation to the common phenotypic changes in class 1. However, the reconstructed mutant strain of *mprA* exhibited a similar, yet significantly different resistance profile for certain stresses, such as SXZ (Supplementary Data 2). The differences between the evolved strains and reconstructed mutant strains might suggest the contribution of multiple mutations, and epistasis among them, to the resistance changes since the evolved strains had several mutations that are potentially involved in resistance acquisition. Moreover, the ability of certain resistance-conferring mutations to impose a different degree of drug resistance on different genetic

backgrounds[52] might reflect the ubiquity of epistatic interactions among genetic alterations. There may have also been a contribution made by non-genetic adaptation, which is difficult to explain by the phenotype-genotype mapping presented in this study. Meanwhile, epigenetic changes e.g., methylation of bacterial DNA, can influence gene expression and/or mutation rates resulting in resistance acquisition[53]. Future studies will likely identify the effect of multiple mutations and non-genetic adaptations on stress resistance, which will enable a better understanding of phenotypic and genotypic constraints on resistance evolution.

A third limitation of our study was the limitation of our transcriptome analysis of the evolved strains to only non-stress conditions. Of course, the expression of some genes would only be induced or suppressed in the presence of a stressor. In such cases, evolution would change such environment-dependent regulatory responses to achieve resistance. However, in this study, we neglected the environment-dependent expression changes and collected gene expression profiles exclusively in the non-stress condition, to compare expression profiles of the evolved strains without environmental-dependent biases. An alternate choice of experimental design would be to collect the gene expression profiles under various stress conditions, to reveal both the environment-specific regulatory responses and their evolution. However, such analysis is costly, and thus, will serve as our study's future scope.

Finally, we identified the decelerated evolution against β-lactams. At present, the mechanism for the observed decelerated evolution remains unclear. We tested several hypotheses to explain this phenomenon, including the effect of fitness cost, negative epistasis, and alternation of mutation frequency by adding β-lactams, and found that they were not sufficient to explain it. However, we further hypothesize that the inhibition of cell wall synthesis by β-lactams addition may disrupt membrane protein synthesis, including OmpF porin. Since the addition of β-lactams reportedly induces bulge formation leading to cell lysis[54], a decrease in *ompF* expression and disruption of OmpF function by a mutation, may not contribute to β-lactam resistance during such a deficient cell wall state. Nevertheless, this phenomenon is clearly interesting, and we expect our observation and testing of several hypotheses will contribute to future studies to unveil the dynamics of antibiotic resistance evolution.

## Methods

**Bacterial strains and growth media.** The insertion sequence (IS)-free *E. coli* strain MDS42[55] was purchased from Scarab Genomics (Scarab Genomics, Madison, Wisconsin, USA) and utilized throughout this study. The use of the IS elements-free strain facilitates reliable resequencing analysis results since the determination of the precise position of IS element insertions is often difficult using short-read sequencers. Although certain essential factors in resistance evolution, including the effects of transposition and horizontal gene transfer, are difficult to analyze in this experimental setup, the use of this strain enables us to identify the precise correspondence between resistance acquisition and mutation fixation. *E. coli* cells were cultured in modified M9 minimal medium containing 17.1 g/L $Na_2HPO_4 \cdot 12H_2O$, 3.0 g/L $KH_2PO_4$, 5.0 g/L NaCl, 2.0 g/L $NH_4Cl$, 5.0 g/L glucose, 14.7 mg/L $CaCl_2 \cdot 2H_2O$, 123.0 mg/L $MgSO_4 \cdot 7H_2O$, 2.8 mg/L $FeSO_4 \cdot 7H_2O$, and 10.0 mg/L thiamine hydrochloride (pH 7.0)[56] plus 15 µg/mL erythromycin. To avoid contamination by other bacteria species, 15 µg/mL erythromycin (approximately 1/10-fold concentration of $IC_{50}$ of *E. coli* MDS42) was added to the medium throughout the experiments. To construct mutant strains, LB medium and Terrific broth (TB) were used. LB medium contained 10 g/L Bacto tryptone, 5 g/L Bacto yeast extract, and 5 g/L NaCl. TB contained 12 g/L Bacto tryptone, 24 g/L Bacto yeast extract, 4 g/L glycerol, 2.32 g/L $KH_2PO_4$, and 12.54 g/L $K_2HPO_4$.

**Laboratory evolution.** Supplementary Data 1 list all of the chemicals used in this study and the solvents in which they were dissolved to prepare stock solutions. Chemicals that did not dissolve in the modified M9 medium were added at a > 20-fold dilution. Cell cultivation, optical density (OD) measurements, and serial dilutions were performed for each chemical using an automated culture system[57] consisting of a Biomek® NX span-8 laboratory automation workstation (Beckman

Coulter, Brea, California, USA) in a clean booth connected to a microplate reader (FilterMax F5; Molecular Devices, San Jose, California, USA), a shaker incubator (STX44; Liconic, Mauren, Liechtenstein), and a microplate hotel (LPX220, Liconic). A movie of the automated culture system is available on YouTube (https://www.youtube.com/watch?v=4k6qCN7ppsk).

Before laboratory evolution, the MDS42 strain was cultivated in a modified M9 medium without chemicals used for laboratory evolution for 96 h (~90 generations) to acclimatize them to the modified M9 medium. After 96-h cultivation, a glycerol stock of the cells was stored as the parent strain at −80 °C for further experiments. The specific growth rate of the parent strain in the modified M9 medium was 0.25 (1/h), which is sufficient for the laboratory evolution setup. Six independent culture lines were evolved in parallel for each chemical. These cultures occurred in 384-well microplates containing 45 µL modified M9 medium per well and a $2^{0.25}$-fold chemical gradient in 22 dilutions. To initiate laboratory evolution, MDS42 cells were inoculated from the frozen glycerol stock into the modified M9 medium and cultivated for 24 h at 34 °C with shaking. The $OD_{620}$ values of the precultures were measured using the automated culture system, and precultured cells, calculated to have initial $OD_{620}$ values of 0.00015, were inoculated into each well (5 µL of diluted overnight culture into 45 µL of medium per well) of the 384-well microplates and cultivated with agitation at 300 rotations/min at 34 °C. Every 24 h of cultivation, cell growth was monitored by measuring the $OD_{620}$ of each well. A well with an $OD_{620} > 0.09$ indicated cell growth. The automated culture system selected the defined well with the highest chemical concentration in which cells could grow; the cells in the selected well were diluted to an $OD_{620}$ of 0.00015 and transferred to fresh plates containing fresh medium and chemical gradients. During and after laboratory evolution, cells were stored in glycerol stocks at −80 °C. We also isolated a single clone on a modified M9 agar plate without chemicals to use for laboratory evolution from the endpoint culture. We have further confirmed that the $IC_{50}$ of the isolated clone was nearly identical to that of the corresponding population in the endpoint culture, where the mean of $IC_{50}$ (isolated clone) − $IC_{50}$ (endpoint culture) was 0.18 ± 0.22 (95% confidence interval). The isolated single clones were used as the evolved strains for further analysis.

**Total RNA purification.** The total cellular RNA was isolated as follows. Cells were inoculated from the frozen glycerol stock into 96-well microplates containing 200 µL of modified M9 medium without chemicals and cultivated for 24 h at 34 °C with agitation at 900 rotations/min. The precultures were diluted to an $OD_{620}$ of $3.3 \times 10^{-5}$–$4.0 \times 10^{-4}$ into 200 µL of fresh modified M9 medium in 96-well microplates and cultured. Cell growth was monitored by measuring the $OD_{600}$ of each well using the 1420 ARVO microplate reader (PerkinElmer Inc., Waltham, Massachusetts, USA). After 12 to 21 h incubation, the cultures exhibiting exponential growth ($OD_{600}$ in the 0.072–0.135 range) were selected and treated immediately by adding an equal volume of ice-cold ethanol containing 10% (w/v) phenol to stabilize the cellular RNA. The cells were harvested by centrifugation at $20,000 \times g$ at 4 °C for 5 min, and the pelleted cells were stored at −80 °C before RNA extraction. Total cellular RNA was isolated using the RNeasy Mini kit (Qiagen, Hilden, Germany) according to the manufacturer's instructions. The total RNA was treated with DNase I at room temperature for 15 min and the purified RNA samples were stored at −80 °C until the microarray experiments were performed.

**Transcriptome analysis using microarrays.** Microarray experiments were performed as in previous study[15] using a custom-designed Agilent 8 × 60 K array for *E. coli* W3110 that included 12 probes for each gene. Briefly, 100 ng of each purified total RNA sample was labeled using the Low Input Quick Amp WT Labeling kit (Agilent Technologies, Santa Clara, California, USA) with Cyanine3 (Cy3) according to the manufacturer's instructions. Cy3-labeled cRNAs were fragmented and hybridized to the microarray for 17 h at 65 °C in a hybridization oven (Agilent Technologies); the microarray was then washed and scanned according to the manufacturer's instructions. Microarray image analysis was performed using Feature Extraction version 10.7.3.1 (Agilent Technologies), and expression levels were normalized using the quantile normalization method[58]. The microarray data have been submitted to the National Center for Biotechnology Information's Gene Expression Omnibus functional genomics data repository under accession number GSE137348.

**qRT-PCR.** mRNA was quantified using the CFX96 Real-Time PCR Detection System (Bio-Rad, Hercules, California, USA). qRT-PCR analysis was performed using the following primer sets: (F) 5′-CGACCTGTTAGACGCTGATT-3′ and (R) 5′-GTTCAGCGGTAACACGGATT-3′ (*gapA*), (F) 5′-CGCGCTTATCGTGAAG AGGC-3′ and (R) 5′-GTGCCGCTGTCGGTCAGTAA-3′ (*tnaA*), (F) 5′-GGCAATGGCGACATGACCTA-3′ and (R) 5′-GCGCCTTCAGAGTTGTTAC C-3′ (*ompF*), (F) 5′-GGCAAGCACCATCAGACTTA-3′ and (R) 5′-CCAGACCTGAGCTGCGTTGT-3′ (*sodB*). A 50 ng RNA sample was used for each RT-PCR with each primer pair using the iTaq Universal SYBR Green One-Step kit (Bio-Rad) according to the manufacturer's instructions. The target gene transcripts were normalized to the reference gene transcript (*gapA*) from the same RNA sample. Each gene was analyzed using RNA isolated from three independent samples. The cycle threshold (CT) for each sample was generated according to the procedures described in the CFX96 Real-Time PCR Detection System user guide.

**Preparation of genomic DNA.** Stock strains were inoculated in 5 mL of modified M9 medium without chemicals in test tubes for 24 h at 34 °C and 150 rpm using water bath shakers (Personal-11, Taitec Co., Nagoya, Japan). Next, 300 µg/mL of rifampicin (RFP) was subsequently added, and the culture was continued for 3 h to inhibit the initiation of DNA replication. The cells were collected by centrifugation at 25 °C and $20,000 \times g$ for 5 min and the pelleted cells were stored at −80 °C before genomic DNA purification. Genomic DNA was isolated and purified using a DNeasy Blood & Tissue kit (Qiagen) in accordance with the manufacturer's instructions. The quantity and purity of the genomic DNA were determined by measuring the absorbance at 260 nm and calculating the ratio of absorbance at 260 and 280 nm ($A_{260/280}$) using a NanoDrop ND-2000 spectrophotometer, respectively. The $A_{260/280}$ values of all the samples were confirmed to be greater than 1.7. The purified genomic DNAs were stored at −30 °C before use.

**Genome sequence analysis using Illumina HiSeq system.** Genome sequence analyses were performed using the Illumina HiSeq System following a previous study[59]. A 150-bp paired-end library was generated according to the Illumina protocol and sequenced using Illumina HiSeq (Illumina, San Diego, California, USA). In this study, 192 samples with different barcodes were mixed and sequenced, which resulted in ~140-fold coverage on average. The potential nucleotide differences were validated using BRESEQ[60].

**Introduction of identified mutations into the parent strain.** To construct mutant strains, the identified mutations were introduced into the parental strain by pORTMAGE[38]. pORTMAGE-4 was a gift from Dr. Csaba Pál (Addgene plasmid # 72679; http://n2t.net/addgene:72679; RRID:Addgene_72679). The introduced mutations and the DNA oligonucleotides used in this study are listed in Supplementary Data 3. MDS42 strain harboring pORTMAGE-4 plasmid was inoculated into 5 mL LB media in the presence of 20 µg/mL chloramphenicol and incubated at 30 °C and 150 rpm. Overnight cultures were diluted 1:100 in chloramphenicol supplemented LB media and grown at 30 °C with agitation at 1000 rotations/min. After 1.5 h incubation, the exponentially growing cultures ($OD_{600} = 0.4$–0.6) were further incubated at 42 °C for 5 min at 1000 rotations/min to induce λ Red protein expression. Cells were then immediately chilled on ice for 10 min. Electrocompetent cells were prepared by washing and pelleting twice in ice-cold sterile MilliQ water. Each MAGE oligo (90-mer oligonucleotide containing the desired mutation and four phosphorothioated bases at the 5′ termini) at 2.5 µM final concentration was introduced by electroporation. For gene inactivation, with the exception of *acrR*, NheI site containing the TAG stop codon was introduced immediately downstream of its start codon and one base was inserted to introduce a frameshift mutation. To construct the *acrR* mutant strain, a NheI site was introduced 24 bp downstream of the start codon. After electroporation, 1 mL TB medium was added, and cells were incubated at 30 °C for 1 h. At this point, cells were subjected to additional MAGE cycles. After the fourth MAGE cycle, cells were diluted 1:1000 and plated onto LB media. After 18 h incubation at 30 °C, single colonies were picked, and corresponding genomic regions were amplified by PCR and verified by Sanger sequencing or NheI digestion to select the desired mutant strains. The selected mutants were further cultivated in TB medium to eliminate the pORTMAGE-4 plasmid. The correctness of the constructed mutants was further confirmed by Sanger sequencing.

**Population analysis profile.** PAP was conducted as previous study[27] with specific modifications. Cells were cultured overnight in the modified M9 medium and $10^{-1}$ to $10^{-6}$ dilutions were subsequently prepared. Next, 5 µL from each dilution and the undiluted culture was dropped on the modified M9 agar plates with increasing amounts of antibiotics (twofold increments). Plates were incubated at 34 °C for 48 h, and colonies were counted to determine the frequency of bacteria growth at each antibiotic concentration. The highest non-inhibitory concentrations were defined as the highest concentration that did not affect the colony-forming unit (cfu) count at least 50% of those that grew on an antibiotic-free plate. A mutant strain was classified as heteroresistant if subpopulations at frequencies of at least $1 \times 10^{-7}$ grew at antibiotic concentrations at least eightfold higher than the highest non-inhibitory concentration of the wild-type. Meanwhile, a mutant strain was classified as genetically resistant if the highest non-inhibitory concentration was at least eightfold higher than that of the wild-type strain.

**Determination of mutation frequencies.** Mutation frequencies were determined as the previous study[61]. The MDS42 strain was first inoculated in the modified M9 medium at 34 °C with shaking (150 rpm). The overnight culture was transferred into 5 mL fresh modified M9 medium and incubated until $OD_{600}$ of the culture reached 0.5. Grows exhibiting exponential growth ($10^8$ cells/mL) were further incubated without drugs or with $IC_{50}$ of antibiotics (CBPC, CMZ, CP, NFLX, or TET) for 4 h at 34 °C with shaking (150 rpm). Subsequently, 2.5 mL of these cultures were centrifuged for 5 min at 10,000 rpm. The pellet was resuspended in 5 mL of fresh modified M9 medium and incubated overnight at 34 °C with shaking. Viable cells were determined by plating appropriate dilutions onto the modified M9 medium agar plates. Mutation frequencies were calculated as the number of colonies growing on rifampicin (100 mg/L) plates per viable colony. Three independent experiments were performed for each condition.

**Prediction and gene selection using a random forest model**. A random forest model was constructed to predict the relative $IC_{50}$s for each of the 47 stresses from the 4492 gene expression levels for all 192 evolved strains. Here, the scikit-learn implementation of the random forest regressor was used. The $\log_{10}$-transformed gene expression levels were used for prediction. Since the relative $IC_{50}$ values had different scales, we normalized the data between stresses by multiplying 1, 0.5, or 0.25 depending on the maximum fold changes of the $IC_{50}$s for the evolved strains against the corresponding chemicals. This normalization corresponds to the chemical gradient dilution steps used for the experimental determination of the $IC_{50}$s. To avoid overfitting, we applied a grid search over the number of trees (16 values between 10 and 40) and the max depth of each tree (60 values between 20 and 1200), using a fourfold cross-validation method. The set of hyperparameters that provided the lowest mean squared prediction error, (number of trees, max depth) = (300, 18), was selected for further analysis. Using the optimal hyperparameters, we trained the random forest on the whole dataset to extract the ranking of relative gene importance for $IC_{50}$ prediction. The relative gene importance was evaluated by the decrease in the mean squared prediction error at each branch of the tree through the feature_importance attribute of the RandomForestRegressor function. The 213 genes which had high-feature importance deviating from the trend of exponential decay were selected for supervised PCA (Supplementary Fig. 5).

**Supervised PCA and hierarchical clustering**. The expression profiles of the 213 genes, selected through the random forest model, were used for PCA (supervised PCA). Hierarchical clustering was applied to the supervised PCA space using Ward's method. The number of classes was determined by the elbow method using class dissimilarity ($W_n$) as a criterion. $W_n$ was defined as $W_n = \sum_k \sum_{i \in c_k} \left| r_i - \mu_{c_k} \right|^2$, where $k$, $c_k$, $\mu_{c_k}$ is the class's index, the set of elements for each class, and class $k$'s centroid, respectively, and $r_i$, $i$ represents the location of each strain, and its index, respectively. For each number of classes $n$, we calculated $W_n$, and its derivative ($W_n - W_{n-1}$) in the resistance space and searched for the number of classes where the derivative of $W_n$ sharply decreased. As a result, the optimal number of classes was determined to be 15. The results of hierarchical clustering including all 15 classes are given in Supplementary Fig. 5. Note, when we varied the number of genes for supervised PCA, the minimum value of $W_{15}$ in the resistance space was observed when the number of genes was 213, which accounts for the precise number of genes used for clustering in the supervised PCA space (Supplementary Fig. 5).

**Linear discriminant analysis**. To investigate the representative gene expression levels for each class, we applied LDA to the transcriptome data. LDA was performed by using the LinearDiscriminantAnalysis function from the scikit-learn package. The strains were assigned binary labels for LDA: one for the strains which belonged to the class of interest, and zero for the other strains. To extract the important genes that characterized each class, we looked for the top-weighted genes in each LDA axis, which corresponded to the genes which contributed to the decision boundary for the binary labeled strains. We further selected the genes that had more than a twofold change in gene expression compared with the parent strain, within the top-weighted genes in the LDA axis.

**Comparing class dissimilarities**. To evaluate how accurately the neighboring relationship in the resistance space was conserved in the supervised PCA space, we calculated $W_{15}$ in the resistance space based on the clustering results in the resistance space, supervised PCA space, mutation space, and the full expression space. For the resistance space, hierarchical clustering was applied based on the 47 relative $IC_{50}$s, to cluster the 192 strains to 15 classes. For the mutation space, hierarchical clustering was applied based on the one-hot encoding which reflects the information of the presence of a mutation. For the expression space, hierarchical clustering was applied to the whole 4492-dimension gene expression space. To construct a baseline, the class dissimilarity was calculated for randomly clustered classes in the resistance space as well. The results are given in Supplementary Fig. 6. We also performed the same procedure for $W_{15}$ in the expression space to determine whether the topological relationships in the expression space correspond with that in the supervised PCA space and resistance space.

**Estimating the ratio of beneficial mutations**. To estimate the ratio of beneficial mutations of the evolved strains, we used the ratio of nonsynonymous to synonymous mutations per site ($dN/dS$) which was 5.26 for the current experiment. Here, we followed the procedure introduced in the supplement of Tenaillon et al.[29,30]. The ratio of beneficial mutations $y$ can be calculated under the assumption that $dN/dS$ should be 1.0 under strict neutrality. This leads to our estimation of $y = (5.26 - 1.0)/5.26 = 0.810$.

**Quantification and statistical analysis**. To determine $IC_{50}$s, serial dilutions of each chemical were prepared in 384-well microplates using the modified M9 medium with $2^{0.25}$-, $2^{0.5}$-, or 2-fold chemical gradients in 22 dilution steps. The chemical gradients depended on the maximum fold changes of the $IC_{50}$s for the evolved strains against the corresponding chemicals. Culture conditions for $IC_{50}$

determination were the same as for laboratory evolution. After 24 h cultivation in the 384-well microplates containing serially diluted chemicals, the $OD_{620}$ of the cultures was measured. Note, this $IC_{50}$ measurement using $OD_{620}$ can be affected by changes in cell size among the evolved strains. To investigate this possibility, we quantified the cell size of all 192 evolved strains cultivated without the stressor and found that no evolved strains showed significantly elongated cell shape in the absence of stressors. We present the measured cell sizes for all 192 evolved strains and the parent strain in Supplementary Fig. 11 (see also "Quantifying single-cell sizes of the evolved strains" below). To obtain the $IC_{50}$ values, the $OD_{620}$ values for the dose-response series were fitted to the following sigmoidal model: $f(x) = \frac{a}{1 + \exp[b(\log_2 x - \log_2 IC_{50})]} + c$, where $x$ and $f(x)$ represent the concentration of antibiotics and observed $OD_{620}$ values, respectively, and $a$, $b$, $c$, and $IC_{50}$ are fitting parameters. The fitting was performed using optimize.curve_fit in the SciPy package[62]. The relative $IC_{50}$ values were computed by comparing the $IC_{50}$ of each evolved strain to the mean of 13 independent measurements of the MDS42 parent strain.

Cross-resistance and collateral sensitivities of stresses were investigated through the Mann–Whitney $U$ test. To detect collateral relationships between stress A and B, we compared the $IC_{50}$ values of stress B of the four strains that evolved in stress A to that of the parent strain (13 independent replicas). The $P$ values were obtained using "wilcox.test" (correct=F, exact=T) in the stats package of R software. We further applied the Benjamini-Hochberg FDR control to these $p$ values so that FDR < 5%.

**Quantifying single-cell sizes of the evolved strains**. To evaluate the cell sizes of the parent and evolved strains, we acquired single-cell images through an inverted microscope BX53 (Olympus, Tokyo, Japan). Single-cell sizes were quantified using a custom ImageJ based code utilizing the "Threshold" and "Analyze Particles" functions. The mean and standard deviation of the cell size of approximately 160 cells for each evolved strain and the parent strain are provided in Supplementary Fig. 11.

**Reporting summary**. Further information on research design is available in the Nature Research Reporting Summary linked to this article.

## Data availability

All microarray data are available in the National Center for Biotechnology Information's Gene Expression Omnibus functional genomics data repository under accession number GSE137348. The raw sequence data of genome sequence analyses are available in the DDBJ Sequence Read Archive under the accession number DRA006396. All relevant data in this study are available from the corresponding authors upon reasonable request. Source data are provided with this paper.

## Code availability

Our custom code for supervised PCA and hierarchical clustering is available at https://github.com/jiwasawa/evolved-strain-analysis/[63]. The custom code is written in Python 3.6, relying on scipy, numpy, pandas, and scikit_learn. Any additional code can be requested from the corresponding author.

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

## Acknowledgements

We thank members of the Furusawa laboratory for fruitful discussions. This work was supported in part by the RIKEN SPDR Program, Grant-in-Aid for the Japan Society for the Promotion of Science (JSPS) Fellows (18J21942), Grant-in-Aid for Early-Career Scientists [18K14688], the Platform for Advanced Genome Science (16H06279), Grant-in-Aid for Scientific Research (S) (15H05746, 19H05626) from JSPS, Grant-in-Aid for Scientific Research on Innovative Areas (17H06389) from Ministry of Education, Culture, Sports, Science, and Technology, and JST-ERATO (JPMJER1902).

## Author contributions

Conceptualization: T.M., J.I., and C.F.; methodology: T.M., J.I., and C.F.; software: J. I., K.T., and C.F.; formal analysis: J.I. and C.F.; investigation: T.M., H.K., N.S., M.K., T.H., and A.S.; supervision: C.F; writing—original draft: T.M. and J.I.; writing–reviewing and editing: C.F.; funding acquisition: T.M., J.I., and C.F.; project administration: C.F.

## Competing interests

The authors declare no competing interests.
