## [Peer Review File · Nature Communications]

Reviewers' Comments:

Reviewer #1:

Remarks to the Author:

This is a huge work with a large dataset generated, but the conclusions reached do not match the expectations. There is a great potential in the paper, but the results are not particularly novel. Specifically, in a previous work performed by the same group, Horinouchi and colleagues initiated high-throughput laboratory evolution under 11 different conditions of environmental stresses and studied evolutionary trade-offs (cross resistance/ collateral sensitivity). Similarly to the current manuscript, Horinouchi et al. explored the underlying molecular mechanisms by analysing genomic expression and genomic mutations in the evolved lineages (Horinouchi et al, Sci Rep. 2017). General patterns behind stress adaptation and cross protection have also been investigated in detail by Dragosits et al. (Mol Syst Biol, 2013) using very similar goals, experimental setups, and bacterial strains.

In the introduction, the authors claim that the evolutionary constraints are crucial for predicting and controlling the evolution of antibiotic resistance: unfortunately, however, the manuscript does not directly address this issue. The authors have found several interesting examples of cross-resistance between pairs of antibiotic and nonantibiotic stressors, but the practical or conceptual importance of these findings are unclear. I believe the authors should attempt to focus on specific examples, and explore them in more detail.

Specific comment

Some of the main conclusions are not particularly novel, e.g: "Overall, our results indicate that the effects of chemical uptake and efflux are major mechanisms for cross-resistance." This has been shown in numerous prior studies (Dam et al., 2018, Wales et al, 2015, Maltas et al, 2019)

The authors have applied as many as 95 antibacterial chemicals including inhibitors of cell wall synthesis, protein synthesis, DNA replication, RNA polymerase, several metabolic pathways, chelators, heavy metals, etc., but the choice of these very heterogeneous group of drugs, or the biological relevance of cross-resistance are unclear.

In the paragraph describing collateral sensitivity, the authors mention that in the reconstructed *sohA* mutant there is a decrease in the expression of *sodB* as it is degraded by the *YhaV* toxin. I assume the authors were referring to the *prfF* and *yhaV* toxin-antitoxin system, but the lack of background information provided and the gene names used (*sohA/prfF*) make it and difficult to understand this section.

The authors found that populations evolving under various stresses acquired higher resistance to β -lactams than strains which were directly selected under β -lactam inhibitor stress. This is an intriguing and surprising result. The explanation, however, is rather sloppy. Why is it that *ompF* mutations are less likely to emerge under β -lactam antibiotic stress?

The subtitle "Decelerated evolution against cell wall synthesis inhibitors" is somewhat misleading, as it focuses on β -lactam only.

Reviewer #2:

Remarks to the Author:

In the present work, Maeda et al perform high-throughput laboratory evolution experiments in *E. coli*, evolving resistance to 95 diverse stressors. The authors perform phenotypic resistance profiling,

whole-genome sequencing and expression profiling assays on the evolved strains and identify clusters of evolved phenotypes by applying machine learning analyses to the gene expression profiles. The authors use these computationally derived clusters to draw conclusions on potential mechanisms for cross-resistance and collateral sensitivity. Overall, the authors present a very impressive body of work that would be of interest to the broad readership of Nature Communications. However, several aspects of the manuscript appear weakly justified and require revision.

(1) The authors describe their analyses as “interpretable machine learning”. This is an over-sell and simply inaccurate, as no biological conclusions were achieved directly from the machine learning model. More precisely, the authors used machine learning to cluster the gene expression profiles and performed feature selection on the genes. These activities do not satisfy the definition of interpretable machine learning.

(2) While the data collection and characterization are laudable, it is not immediately clear to me what novel biological insights were gained in this work. It appears that all of the biological conclusions are dependent on clusters of resistance phenotypes. It is very unclear how unique or optimal these clusters should be. One of the most interesting observations from this work is depicted in Fig. Sf, where W_n appears least for IC50 and Supervised PCA. The authors use this to defend the idea that the clusters drawn from Supervised PCA closely represent resistance phenotypes (defined by the IC50s), but if W_n is smaller using the IC50s alone, why not cluster by IC50 directly?

(3) Relatedly, why does Fig. 2 only depict 11 of the 15 clusters? If the other 4 clusters are not important, why were there 15 clusters to begin with? These 4 clusters seem arbitrarily omitted, and it's unclear how distant those clusters are from the 11 that are shown.

(4) Also, why do the authors not show a plot of the PCA to visualize how similar/different the gene expression profiles are from one another? One would expect that these 15 clusters would be distinctly identifiable by inspection.

(5) Moreover, gene expression data was collected for all clones in only one growth state/culture condition. What is the rationale for this? One would expect that some genes would only be induced in the presence of a stressor, and it seems highly possible that the authors might be completely missing a relevant response by not looking at gene expression profiles in the presence of stress.

(6) There are several instances where the authors jump from high-level descriptions of the data to very specific examples, without explicitly communicating how representative the examples are. For instance, in Figure 3, why did the authors choose the pairs CBPC/AZT and TET/B.Cl.Ala vs other pairs? It would seem obvious that one can arbitrarily find highly- or lowly-correlated pairs.

(7) For Figures 3D and S2a, why are values on the self-self diagonal zero (white)? One would expect those to be identically high (one).

(8) It is well appreciated that transport is a major resistance mechanism that induces non-specific protection from multiple stresses. Transport-related mutations appear over-represented in the authors' analyses and I wonder how much of the correlation depicted in Figure 3C is just due to transport-dependent resistance? In other words, do non-transport related mutations also have such strong correlations in phenotype between the evolved strains and constructed mutants?

(9) The authors describe differences in acquired resistance as “decelerated evolution”. This seems imprecise, as the rate of resistance acquisition is expected to be set by the mutation rate and purifying selection, and the present work addresses neither of these. Instead, the authors are really just describing how different resistance alleles can confer different amounts of protection. Resistance alleles acquired by lab evolution are very sensitive to choice of selection methodology. Could this “deceleration” just be an artifact of their evolution scheme? In fact, ~300 total generations is a

modest length for a lab evolution experiment.

(10) In their abstract, the authors state that they report a “novel constraint that leads to decelerated evolution”, but it is not at all clear what this novel constraint is. In fact, line 292 states “At present, the mechanism for the decelerated evolution against beta-lactams remains unclear.” This manuscript would overall benefit from a formal treatment on these “evolutionary constraints” as referenced in the title and abstract.

Reviewer #3:

Remarks to the Author:

Manuscript titled “High-throughput laboratory evolution reveals evolutionary constraints in *Escherichia coli*” describes experimental evolution of *Escherichia coli* MDS42 strain. Tomoya Maeda and co-workers evolved cells in 384-well plates, where each plate contained a modified M9 medium, glucose and an increasing concentration of a particular antimicrobial compound. They used automatic experimental system, developed previously by the Chikara Furusawa’s group, to evolve populations in 95 different antimicrobial compounds with 6 replicates, which means that they run in parallel a striking number of 576 plates. They experimentally evolved populations for 27 days and in evolved strains determined the transcriptome, whole-genome sequence and the half-maximal inhibitory concentration (IC50) of the antimicrobial compound that the strain was incubating with. Then they tested if specific mutations (identified with DNA sequencing) were responsible for the increased IC50 (relative to the parent), a phenotype that was measured in majority of evolved strains. Using an elegant method developed by Csaba Pal’s group they introduced these specific mutations into ancestor (parental) strain. They measured in many mutated parental strains a similar IC50 shift as measured in the evolved strain, thus demonstrating that specific mutations could be responsible for the increased IC50.

Studying genotype-phenotype mapping is an important research area, especially when a studied phenotype is a major health concern, which antimicrobial resistance (AMR) is. The high-throughput experimental approach that the group has developed is spectacular and the list of tested compounds is very well thought. I find evidence that specific mutations are shifting IC50 quite compelling.

However, certain aspects of the experimental design require an explanation. There is also a worrying lack of details regarding why they excluded so much data. I also believe that text and figures should be clearer and the entire manuscript should be better balanced with the existing knowledge on AMR. I hope my specific comments can improve the manuscript, because this solid and important work must come out and will be of interest to many AMR researchers.

1. Experimental design:

1.1 I wonder why you decided to use *E. coli* MDS42 strain. You probably wanted to constrain evolution to single-point mutations (thus avoid insertion sequences), and you perhaps wanted to work with evolved strains that experienced lower number of mutational events (given that MDS42 is promoted as anti-mutator strain). Whatever your motivation you need to explain it, and you have to discuss how this unusual strain might impact the generalisation of your results.

1.2 In the past I did experiments with MDS42 strain, which I have not yet published. I can say that MDS42 has a poor growth in Davis minimal medium (DM) with glucose as the only source of carbon and energy (recipe for DM can be found in Lenski et al. 1991, *American Naturalist*). I have not tried to grow MDS42 in M9, but I wonder, is poor growth in a standard M9 medium the reason why you used a modified M9? I believe you should explain why you modified the medium.

1.3 To the best of knowledge, I cannot think of why you added sub-lethal concentration of erythromycin (15µm/ml) into modified M9 (page 18/line 367). I believe erythromycin is not effective against fungi or viruses (which could potentially contaminate your experiment). If you really used it, then you have to say why and discuss how is this relevant for the interpretation of your results. At the moment you are not discussing presence of erythromycin at all.

1.4 You grew cells in 5 g of glucose per L (page 18/line 365). I expect that MDS42 will never use all that glucose in 24 hours, especially not on 34°C. In such conditions cells incubated with different antimicrobial compounds could grow differently not only in numbers but also in size. Your main measurement is OD, which is sensitive to cell's size. How much do you think is your IC50 measurement sensitive to cell's size? Is it possible that at least some variation in observed IC50 shifts could be down to variation in cell's size in the parent versus evolved strain?

2. Data exclusion:

2.1. I wonder why and how you selected 192 evolved strains (page 5/line 49). For instance, I know for sure that you can experimentally evolve susceptible *E. coli* cells in non-lethal concentrations of trimethoprim (which is one of the environment you excluded).

2.2. Also, why you analysed only 4 independent cultures from the same environment, because you started with six. In the text you only mention six cultures in the first result paragraph and on a few places in the supplementary material. I am convinced that you had good reasons to do it, but you have to explain them properly.

2.3. On page 5 you mention GAH, NQO and MMC environments, where you detected higher number of mutations. Why you decided not to follow up this potentially important environments?

3. Clarity

3.1. It looks to me that 300 generations (page 5/line 45) is more like a rough estimate over the entire experiment. I think you have to explain how you come up with this number. As I understand, you transferred 5 µl of diluted 24-hours culture into 45 µl of a fresh medium, where the 5 µl inoculum were prepared by diluting an overnight culture to OD620 of 0.0003. Surely ODs of overnight cultures were not the same across environments, which means that the number of generations per day could vary not only per environment but also across the experimental evolution. Usually, if you dilute cultures 100-fold, the number of generations per day is 6.64.

3.2. There is some inconsistency when you reported the number of control replicates that were compared to evolved strains. You started with six independent parental lines (page 4/line 44), but then you say on page 5 (line 50) "...plus a control without any stress". Then on page 26 (lines 564 & 568) you stated that "...the mean of 13 independent replicas of the parent strain".

3.3. Sentence that starts on page 5/line 62 should be moved to a more appropriate place, probably where transcriptome data is first introduced.

3.4. On page 5 (line 67) you need to be more specific. When you say "about 80%", is this 80% out of 47 or 95 environments. You need to report the number of strains with some measure of variance. And what is the variation in the number of mutations among strains evolved in the same environment?

3.5. On page 5 (line 70) you mentioned only strains evolved in GAH, but what about NQO and MMC. It would be better if you tell us the mean with CI for all three environments.

3.6. Figure 1c (page 6/line 74) is slightly misleading. First column probably shows 21 "synonymous" mutations (it is hard to be sure given the small size of the panel). So, probably, "other point mutations", "duplications", "indels" and "large deletions" (next four columns) are all non-synonymous mutations? Perhaps you could show only four columns (four types) and have two colours (synonymous & non-synonymous).

3.7. I do not understand where is 80% coming from (page 6/line 76). Please explain the number.

3.8. I do not think you have enough evidence to describe stressors that you included as non-mutagens (page 6/ line 74). As far as I know at least 6-mercaptopurine monohydrate, acriflavine, hydrogen peroxide, nitrofurantoin were shown to be mutagenic in bacteria (and this is probably not an exhaustive list).

3.9. You need to properly define what you mean by commonly mutated gene (page 7/ line 114).

3.10. On page 10 (line 173) you mentioned 64 representative mutations. Are these mutations located only in those 25 genes? Have you introduced mutations also from the rest of 188 mutated genes?

3.11. On page 11 / line 196 you reported results from the erythromycin environment. It is important to say something about erythromycin being present in your medium.

3.12. On page 15 / line 286 you reported very marginal p value that I would not necessarily trust. You also do not mention statistical test or number of observations. I believe presenting N, measure of

variation and identity of the statistical test is a gold standard for each report.

3.13. Regarding the decelerated evolution, I think you should discuss the potential effect of mutation supply rate on your results. A link between mutation supply rate and adaptation rate as shown in this paper (Salverda et al. *Proceedings of the National Academy of Sciences* 114, 12773-12778, 2017), might be relevant.

3.14. Please, you have to help the reader to better understand so many of your abbreviations. For instance, you mention B.Cl.Ala on page 15 / line 307 (but defined it on page 13 / line 241). Also names on page 17 (line 334) are defined way back in the Result section. I also suggest that you try to use as many standard abbreviations as possible, I believe 5-Fluoroorotic Acid is usually abbreviated as 5-FOA.

3.15. Statement on page 16 / line 324 is probably an overstatement, because you have cases where mutation introduction was not enough (which you acknowledged later on page 17 / line 348), so I would rephrase it.

3.16. On page 19 / line 404 you mentioned that the IC50 of the isolated clone was almost identical. This is important information, so give this statement a proper weight by reporting both means with CI.

3.17. Page 20 / line 413: Did you mean OD600 or OD620.

4. More than just terminology

4.1. I think it is important that you introduce distinctions between heteroresistance, persistence, tolerance and genetic resistance (as defined in Balaban et al. 2019, *Nature Reviews Microbiology*). On short, genetic resistance is an increase in minimal inhibitory concentration (MIC) due to a resistance mutation or a plasmid-borne resistance gene, whereas heteroresistance entails a MIC increase without obtaining genetic resistance (e.g. higher expression of efflux pumps). Tolerance is an increase in the minimum duration to kill 99% of the population (MDK99) and persisters are a subpopulation of dormant tolerant cells with a high MDK99. Also, resistant and heteroresistant cells divide in the presence of the antibiotic, while tolerant cells and persisters do not. I strongly suggest that you consistently use the terminology from this seminal paper, written by AMR pioneers aiming to decrease confusion in AMR research.

4.2. In your study it is not clear what is actually being affected by mutations, but I would say that across all environments cells mostly evolved heteroresistance. For instance, you can enrich the bacterial culture with tolerant cells by pre-exposing them to antibiotics that arrest protein synthesis (Kwan et al. *Antimicrob. Agents Chemother.* 2013), like rifampicin (RFP), tetracycline (TET) and carbonyl cyanide 3-chlorophenylhydrazone (CCCP), all three compounds were also part of your study. Just to make a point, for the rifampicin I am convinced that you have not sequenced the resistant strain, because genetic resistance to rifampicin is possible only via ~79 point mutations within an *rpoB* gene (Garibyan et al., *DNA Repair* 2, 593-608). What you probably evolved in rifampicin environment is a heteroresistant strain with the 1.5-fold increased MIC. Note that Kwan's paper is published in 2013, when tolerance was still confused with heteroresistance.

4.3. My suggestion is that you rewrite your results and discussion by really focusing on novel connections that you found. And whenever you confirmed an already known connection, you should cite a relevant paper. Regarding the tolerance/heteroresistance/resistance dilemma that I just raised, I would double check again all the genes in all the environments and make sure that the right term is used for describing the evolutionary outcome.

4.4. I do not agree with your definition of cross-resistance and collateral sensitivity on page 3 (line 10), because double genetic resistance is a very rare event. I believe acquisition of resistance (and heteroresistance) to a certain drug can be accompanied by higher or lower heteroresistance (or tolerance) to another drug.

4.5. At the end of the discussion you listed other explanations named as "multiple mutations", "epistasis" and "non-genetic adaptation". I am not convinced that they are really an alternative, but whatever your idea is you need to be more specific and give us some references. For instance, epigenetics in AMR has been proposed although it is not in a mainstream (Ghosh et al. *Antimicrob. Agents Chemother.* 64, e02225-02219). Also there is a lot of literature linking epistasis with AMR (see please Wong, A. *Front. Microbiol.* 8, 246-246).

Reply to Reviewer #1

Comment 0: This is a huge work with a large dataset generated, but the conclusions reached do not match the expectations. There is a great potential in the paper, but the results are not particularly novel. Specifically, in a previous work performed by the same group, Horinouchi and colleagues initiated high-throughput laboratory evolution under 11 different conditions of environmental stresses and studied evolutionary trade-offs (cross resistance/ collateral sensitivity. Similarly to the current manuscript, Horinouchi et al. explored the underlying molecular mechanisms by analysing genomic expression and genomic mutations in the evolved lineages (Horinouchi et al, Sci Rep. 2017). General patterns behind stress adaptation and cross protection have also been investigated in detail by Dragosits et al. (Mol Syst Biol, 2013) using very similar goals, experimental setups, and bacterial strains.

Response 0: We would like to thank the reviewer for their constructive suggestions to improve our manuscript. We have revised the manuscript accordingly and have provided our point-by-point responses below.

Comment 1: In the introduction, the authors claim that the evolutionary constraints are crucial for predicting and controlling the evolution of antibiotic resistance: unfortunately, however, the manuscript does not directly address this issue. The authors have found several interesting examples of cross-resistance between pairs of antibiotic and nonantibiotic stressors, but the practical or conceptual importance of these findings are unclear. I believe the authors should attempt to focus on specific examples, and explore them in more detail.

Response 1: We appreciate the reviewer's critical comments. We agree that the present study did not directly address the prediction and control of the evolutionary dynamics. According to the reviewer's suggestion, we thoroughly revised the Introduction, Results, and Discussions to present detailed explanations on our new findings. The findings include new resistance-conferring mutations (e.g. *prfF*, *ycbZ*, and *yhjE*), new cross-resistance/collateral sensitivity relationships (e.g. *rpoS* associated metabolic inhibitor sensitivity and *prfF* associated oxidative stress sensitivity), and decelerated evolution against β -lactam antibiotics. Moreover, based on comments from another reviewer, we also conducted a population analysis profile (PAP) to examine whether the main, or only sub-population of cells, are phenotypically resistant (the latter case represents heteroresistance). Heteroresistance is a common phenomenon of several bacterial species and antibiotic classes in which subpopulations of the main population of susceptible cells show increased resistance (Andersson *et al.*, *Nat. Rev. Microbiol.*, **17**, 479-496 (2019); Balaban *et al.*, *Nat. Rev. Microbiol.*, **17**, 441-448 (2019); Band *et al.*, *Nat. Microbiol.*, **4**, 1627-1635 (2019); Nicoloff *et al.*, *Nat. Microbiol.*, **4**, 504-514 (2019)). The results of the additionally conducted PAP allowed us to identify many

heteroresistance-conferring mutations in novel genes as well as known repressors for multidrug efflux pumps. We also additionally examined the mutation frequencies in the presence of various antibiotics to address the possible involvement of mutation supply rate in the decelerated evolution phenomenon. Hence, although we did not directly focus on predicting and controlling the evolution of antibiotic resistance in the current study, we believe that our new findings and constraints for resistance evolution could provide a basis for further analysis in these areas. To discuss these issues in more detail, we revised the introduction, results and discussion sections.

Revision 1: Introduction (Page 4, Line 32-37)

“To examine whether the whole population or only a subset of cells were phenotypically resistant, we conducted a population analysis profile (PAP). Heteroresistance is a common phenomenon for several bacterial species, and antibiotic classes, in which a subpopulation among susceptible cells exhibits increased resistance^{24–27}. Here, we identified many heteroresistance-conferring mutations in novel genes, as well as known repressors for multidrug efflux pumps.”

Revision 1: Results (Page 18, Line 370-372)

“To address this possibility, we measured mutation frequencies with β -lactams (CBPC and CMZ), or other drugs showing higher β -lactam resistance (CP, NFLX, TET); however, differences were not observed in mutation frequencies (Supplementary Fig. 10).”

Revision 1: Discussion (Page 20, Line 414-419)

“Furthermore, the analysis identified novel genetic mechanisms for the stress resistance, such as CBPC, AZT, and DVAL resistance through *prlF* mutation (Fig. 3e), which is related to the TA system. We also discovered a contribution made by uncharacterized genes to resistance acquisition, such as the *ycbZ* and *yhjE* mutations causing EM, NMNO, EDTA, NVA, and 5-FOA resistances, and PLM, NVA, and DVAL resistances, respectively.”

Comment 2: Some of the main conclusions are not particularly novel, e.g: “Overall, our results indicate that the effects of chemical uptake and efflux are major mechanisms for cross-resistance.” This has been shown in numerous prior studies (Dam et al., 2018, Wales et al, 2015, Maltas et al, 2019).

Response 2: As the reviewer pointed out, the effects of suppressing chemical uptake and activating efflux pumps have been previously reported as major mechanisms causing cross-resistance among antibiotics. However, in this study, by expanding the quantification of cross-resistance/collateral sensitivity network to various stressors including both antibiotics and non-antibiotics, we confirmed the contribution made by altering uptake/efflux activities in the cross-resistance relationships among them. We should stress here that this analysis enabled us to identify novel relationships between

transporters and substrates. For example, we identified a new transporter YhjE involving DL-3-hydroxynorvaline (NVA) and D-valine (DVAL) resistance. Such an analysis of transporter-substrate specificity provided detailed understandings of the mechanism of cross-resistance. We added a paragraph to Discussion to explain our findings related to the expanded cross-resistance and collateral sensitivity network.

Revision 2: Results (Page 13, Line 246-251)

“Antibiotic resistance of *E. coli* can be triggered by the overexpression of efflux systems and decreased production of porin proteins^{6,34,39}. In this study, by expanding the quantification of cross-resistance/collateral sensitivity network to various stressors including antibiotics and non-antibiotics, we confirmed the contribution of altered uptake/efflux activities in the cross-resistance relationships among them (Fig. 2d, Table 1, Supplementary Table 5).”

Comment 3: The authors have applied as many as 95 antibacterial chemicals including inhibitors of cell wall synthesis, protein synthesis, DNA replication, RNA polymerase, several metabolic pathways, chelators, heavy metals, etc., but the choice of these very heterogenous group of drugs, or the biological relevance of cross-resistance are unclear.

Response 3: In this study, we chose heterogenous stressors, including antibiotics with various mechanisms of action, as well as other non-antibiotic toxic chemicals to *E. coli*. The reason for choosing such a variety of stressors was to analyze the expanded cross-resistance/collateral sensitivity network and to elucidate the molecular mechanisms for the resistance acquisitions. As a result, we identified novel cross-resistance/collateral sensitivity relationships and genetic basis for them. We also determined that the resistance acquisitions to various stressors resulted in evolution to similar phenotypes, suggesting constraints in the resistance evolution. These results can serve as a basis for prediction and control of resistance evolution by combinatorial usage of these antibiotics and non-antibiotics stresses, which will be applicable to prevent the spread of antibiotic resistance. It should be stressed that all results can be obtained only through our approach, that is, laboratory evolution with an expanded choice of stressors. To explain the experimental design in more detail, we revised the Introduction and Discussion as follows.

Revision 3: Introduction (Page 3-4, Line 19-25)

“Here, we performed high-throughput laboratory evolution of *Escherichia coli* under 95 heterogeneous stressors (Supplementary Table 1). To analyze the expanded cross-resistance/collateral sensitivity network, including both antibiotic and non-antibiotic stressors, while elucidating the molecular mechanisms associated with resistance acquisition, we chose a variety of antibacterial chemicals, including antibiotics with various mechanisms of action, and non-antibiotic toxic chemicals, against *E. coli*.”

Revision 3: Discussion (Page 18, Line 377-379)

“In this study, we performed laboratory evolution of *E. coli* under various heterogenous stress conditions which allowed us to elucidate the molecular mechanisms associated with resistance acquisition to antibiotics and non-antibiotics stressors.”

Comment 4: In the paragraph describing collateral sensitivity, the authors mention that in the reconstructed *sohA* mutant there is a decrease in the expression of *sodB* as it is degraded by the YhaV toxin. I assume the authors were referring to the *prfF* and *yhaV* toxin-antitoxin system, but the lack of background information provided and the gene names used (*sohA/prfF*) make it and difficult to understand this section.

Response 4: According to the reviewer’s suggestion, we have replaced the gene name *sohA* to *prfF* in the manuscript, figures, and supplementary tables to clarify its relation to the PrfF–YhaV toxin-antitoxin system. To save space, we have provided a single example of one such revision below:

Revision 4: Results (Page 8-9, Line 144-147, etc.)

“For example, all evolved strains in class 1 had mutations in *mprA*, which encodes a repressor for multidrug resistance pump EmrAB, while all strains in class 11 had mutations in *prfF*, which encodes the antitoxin for the PrfF (SohA)-YhaV toxin-antitoxin (TA) system.”

Comment 5: The authors found that populations evolving under various stresses acquired higher resistance to β -lactams than strains which were directly selected under β -lactam inhibitor stress. This is an intriguing and surprising result. The explanation, however, is rather sloppy. Why is it that *ompF* mutations are less likely to emerge under β -lactam antibiotic stress?

Response 5: One possible explanation is that decreased evolution is caused by the difference in mutation frequencies between β -lactams and other stresses. To address this possibility, we measured mutation frequencies under the addition of IC_{50} concentrations of carbenicillin (CBPC), cefmetazole (CMZ) (both β -lactams), chloramphenicol (CP), tetracycline (TET), and norfloxacin (NFLX), as well as in the absence of drugs. The results demonstrated that mutation frequencies between β -lactams and other stresses do not differ (see figure below; Supplementary Fig. 10d in the revised manuscript). This indicated that the change in mutation frequency does not effectively explain the observed decelerated evolution, at least for the evolved strains under CP, TET, and NFLX. At present, the molecular mechanism of this phenomenon remains unclear. We hypothesized that the inhibition of cell wall synthesis by β -lactam addition may disrupt membrane protein synthesis, including that of OmpF porin. Since the addition of β -lactams is known to induce bulge formation leading to cell lysis, a decrease in *ompF* expression and disruption of OmpF function via mutation, may not contribute to

β -lactam resistance during such a deficient cell wall state. The phenomenon is clearly interesting and might contribute to a detailed understanding of β -lactam resistance evolution. Hence, the possible mechanisms, including our hypothesis, should be investigated further.

Revision 5: Results (Page 18, Line 367-372)

“Since a link between mutation supply rate and adaptation rate was reported ⁴⁹, it is also possible that decreased evolution is caused by the difference in mutation frequencies between β -lactams and other stresses. To address this possibility, we measured mutation frequencies with β -lactams (CBPC and CMZ), or other drugs showing higher β -lactam resistance (CP, NFLX, TET); however, differences were not observed in mutation frequencies (Supplementary Fig. 10).”

Revision 5: Discussion (Page 22, Line 461-471)

“Finally, we identified decelerated evolution against β -lactams, however, examination of fitness cost, negative epistasis, and differences in mutation frequency were not sufficient to explain this phenomena. Currently, the associated molecular mechanism remains unclear; however, we hypothesize that the inhibition of cell wall synthesis by β -lactams addition may disrupt membrane protein synthesis, including OmpF porin. Since the addition of β -lactams reportedly induces bulge formation leading to cell lysis ⁵⁴, a decrease in *ompF* expression and disruption of OmpF function by a mutation, may not contribute to β -lactam resistance during such a deficient cell wall state. Nevertheless, this phenomenon is clearly interesting, and might contribute to a detailed understanding of β -lactam resistance evolution, however, the associated mechanisms, including our hypothesis, should be investigated as future works.”

Supplementary Fig. 10d

Mutation frequencies for MDS42 strains under addition of IC₅₀ concentrations of β -lactam stresses (i.e. CBPC, CMZ) and other antibiotics in which the evolved strains acquired high resistance to β -lactam stresses (i.e. CP, NFLX, TET). Error bars represent the standard deviation of three independent experiments.

Comment 6: The subtitle “Decelerated evolution against cell wall synthesis inhibitors” is somewhat misleading, as it focuses on β -lactam only.

Response 6: According to the reviewer’s suggestion, we have revised the subtitle as follows: “Decelerated evolution against β -lactam antibiotics.”

Reply to Reviewer #2

Comment 0: In the present work, Maeda et al perform high-throughput laboratory evolution experiments in *E. coli*, evolving resistance to 95 diverse stressors. The authors perform phenotypic resistance profiling, whole-genome sequencing and expression profiling assays on the evolved strains and identify clusters of evolved phenotypes by applying machine learning analyses to the gene expression profiles. The authors use these computationally derived clusters to draw conclusions on potential mechanisms for cross-resistance and collateral sensitivity. Overall, the authors present a very impressive body of work that would be of interest to the broad readership of Nature Communications. However, several aspects of the manuscript appear weakly justified and require revision.

Response 0: We would like to thank the reviewer for their constructive suggestions to improve our manuscript. We have revised the manuscript accordingly and provided our point-by-point responses below.

Comment 1: The authors describe their analyses as “interpretable machine learning”. This is an over-sell and simply inaccurate, as no biological conclusions were achieved directly from the machine learning model. More precisely, the authors used machine learning to cluster the gene expression profiles and performed feature selection on the genes. These activities do not satisfy the definition of interpretable machine learning.

Response 1: We agree with the reviewer that, in this study, biological conclusions were not achieved directly from our random forest model nor hierarchical clustering in the supervised PCA space. Instead, we have identified the underlying biological mechanisms for each cluster using a simple machine learning method (LDA). The term “interpretable machine learning” has a slightly broad meaning; for example, the investigation of features that lead to the model’s decision is regarded as interpretable machine learning (e.g., Azodi *et al*, *Trends Genet.*, **36**, 442-445 (2020)), and there exist cases where LDA is regarded as interpretable machine learning (e.g., Wu *et al*, *Bioinformatics*, **25**, 1145-1151 (2009)). However, according to the reviewer’s suggestion, we admit that our usage of interpretable machine learning led to a certain degree of confusion and have, therefore, revised the associated text as follows:

Revision 1: Abstract (Page 2)

“Utilizing machine learning techniques, we analyzed the phenotype-genotype data and identified low dimensional phenotypic states among the evolved strains.”

Revision 1: Introduction (Page 4, Line 28-30)

“By analyzing the gene expression-resistance map through machine learning techniques, the emergence of low dimensional phenotypic states was observed in the evolved strains, indicating the existence of evolutionary constraints.”

Revision 1: Discussion (Page 18, Line 379-381)

“Combined with supervised PCA and hierarchical clustering, our high-throughput phenotypic analysis led to the identification of modular phenotypic classes”

Comment 2: While the data collection and characterization are laudable, it is not immediately clear to me what novel biological insights were gained in this work. It appears that all of the biological conclusions are dependent on clusters of resistance phenotypes. It is very unclear how unique or optimal these clusters should be. One of the most interesting observations from this work is depicted in Fig. Sf, where W_n appears least for IC50 and Supervised PCA. The authors use this to defend the idea that the clusters drawn from Supervised PCA closely represent resistance phenotypes (defined by the IC50s), but if W_n is smaller using the IC50s alone, why not cluster by IC50 directly?

Response 2: The aim of the clustering analysis in Fig. 2 was to extract evolved strains with similar resistance phenotypes and to clarify their biological mechanisms of resistance. We determined that clustering in the supervised PCA space successfully represented phenotype clusters, enabling us to elucidate the underlying resistance mechanisms as common expression changes and mutations in each cluster. As the reviewer pointed out, there is an alternative strategy for clustering, i.e., clustering IC₅₀ directly instead of clustering by gene expressions after supervised PCA. However, clustering directly by IC₅₀ makes it difficult to observe associations between phenotypic clusters and biological mechanisms. For example, when we performed clustering by IC₅₀, the evolved strains with high *prf1/yhaV* activity were not always in the same cluster. This “mixing” of the clusters was also observed in the evolved strain with high *mprA* activity. These clustering results were in contrast with the clear correspondence between resistance phenotype and expression changes/mutations shown in Fig. 2, which was obtained using the clustering in the supervised PCA space.

The differences obtained from these two clustering methods is likely due to the smaller effective dimension in the IC₅₀ space compared to the supervised PCA space. In fact, when we performed PCA for the IC₅₀ space, supervised PCA space, and the whole expression space, the number of dimensions corresponding to 90% variance was 18, 36, and 70, respectively. If direct clustering in the IC₅₀ space shows a different structure compared to that in the expression space, it would indicate that IC₅₀ data alone is not sufficient to recover information of the gene expression data, making it difficult to interpret the underlying biological mechanisms for each resistance phenotype.

To further investigate the differences between clustering methods, we compared the class dissimilarity W_{15} in the expression space. The associated data has been added to the revised manuscript as Supplementary Fig. 6b (shown below). Although clustering in the supervised PCA

space represents the clusters of the whole expression space well, the class dissimilarity W_{15} obtained by the clustering in IC_{50} space approaches that of random clustering. It should be stressed that the clustering in the supervised PCA space shows relatively small W_{15} in IC_{50} space, as shown in Supplementary Fig. 3f in the original manuscript (shown below; this figure is renamed to Supplementary Fig. 6a in the revised manuscript). These results indicate that clustering by the supervised PCA space represents the data structure well for both IC_{50} and gene expression spaces. This could be explained by the difference in the number of effective dimensions in the IC_{50} space and supervised PCA space, as described above. This additional result validates our approach using the information in the supervised PCA space for clustering.

We have also calculated W_{15} in the IC_{50} space for the classes in the supervised PCA space constructed by the different number of genes (Supplementary Fig. 5f). Here, the N genes were obtained from the top N important genes from the random forest model. As a result, W_{15} for the classes in the supervised PCA space constructed with 213 genes was assigned the minimum value, indicating that the classes we used for analysis correspond well with the structure in the IC_{50} space.

Supplementary Fig. 5f and Supplementary Fig. 6a, b together show that the supervised PCA space provides adequate mapping between the gene expression and IC_{50} space. To explain our additional analyses, we have modified the manuscript as follows.

Supplementary Fig. 5f, Class dissimilarity W_{15} for the results of hierarchical clustering in the supervised PCA spaces constructed by different number of genes. The genes are sorted in the order of gene importance given by the random forest model.

Supplementary Fig. 6. Accessing the validity of clustering in the supervised PCA space

a, Class dissimilarity W_{15} in the IC_{50} space for the 15 classes defined by hierarchical clustering based on the supervised PCA expression space, IC_{50} space, mutations, and full gene expression space, respectively, are shown. The mean and standard deviation for the class dissimilarity for ten runs of randomly clustered results in the IC_{50} space are also shown. b, Class dissimilarity W_{15} in the 4492 dim. gene expression space for the results of hierarchical clustering in other spaces are shown. The mean and standard deviation for the class dissimilarity for ten runs of randomly clustered results in the gene expression space are also shown.

Revision 2: Results (Page 8, Line 126-134)

“In contrast, direct clustering in the resistance space did not necessarily correspond with characteristic gene expression profiles. When we calculated the class dissimilarity W_n in the gene expression space based on the clustering result in resistance space, the value approached that of random clustering (Supplementary Fig. 6b). This was likely due to the effective degrees of freedom in the resistance space not being sufficient to recover necessary information in the gene expression space, while the supervised PCA space obtained good representations conserving information for both the expression and resistance space (Supplementary Fig. 6).”

Revision 2: Methods (Page 31, Line 678-682)

“Note, when we varied the number of genes for supervised PCA, the minimum value of W_{15} in the resistance space was observed when the number of genes was 213 which accounts for the precise number of genes used for clustering in the supervised PCA space (Supplementary Fig. 5).”

Comment 3: Relatedly, why does Fig. 2 only depict 11 of the 15 clusters? If the other 4 clusters are not important, why were there 15 clusters to begin with? These 4 clusters seem arbitrarily omitted, and its unclear how distant those clusters are from the 11 that are shown.

Response 3: The other four clusters contain three singletons (a cluster with only a single strain), while the remaining cluster had gene expression levels close to that of the parent MDS42 strain. In Fig. 2, we have omitted these four clusters due to the visibility, as we note in the legend of Fig.2. According to the reviewer’s comment, to present the distance of these other clusters, we added Supplementary Fig. 4 to the revised manuscript, which presents PCA and tSNE plots of the expression profiles in 192 evolved strains in the supervised PCA space. As we can see in Supplementary Fig. 4c,d,f, the omitted cluster 12 shows gene expression profiles close to that of the parent strain.

Supplementary Fig. 4. PCA and tSNE plot of the supervised PCA space

a, b, Distribution of the 192 evolved strains and the parent strain in the supervised PCA space. The four principal components are shown and colors denote the evolved environment for each strain. c, d, Data is the same as in a, b is plotted. The colors denote the class defined by hierarchical clustering. e, The tSNE plot for the distribution of the evolved strains in the 36 dimension supervised PCA space. The colors denote the evolved environment. f, Data is the same as in e. Colors denote the classes defined by hierarchical clustering.

Comment 4: Also, why do the authors not show a plot of the PCA to visualize how similar/different the gene expression profiles are from one another? One would expect that these 15 clusters would be distinctly identifiable by inspection.

Response 4: According to the reviewer's suggestion, we plotted the first 4 dimensions of the supervised PCA space and added this data as Supplementary Fig. 4 a-d. Moreover, to give an intuition of the whole 36 dimension supervised PCA space, used for clustering, we also show a t-SNE plot of the whole 36 dimension supervised PCA space in Supplementary Fig. 4 e, f. Here, 36 dimensions of the supervised PCA space accounts for 90% of the total variance.

Supplementary Fig. 4. PCA and tSNE plot of the supervised PCA space

a, b, Distribution of the 192 evolved strains and the parent strain in the supervised PCA space. The four principal components are shown and colors denote the evolved environment for each strain. c, d, Data is the same as in a, b is plotted. The colors denote the class defined by hierarchical clustering. e, The tSNE plot for the distribution of the evolved strains in the 36 dimension supervised

PCA space. The colors denote the evolved environment. f, Data is the same as in e. Colors denote the classes defined by hierarchical clustering.

Comment 5: Moreover, gene expression data was collected for all clones in only one growth state/culture condition. What is the rationale for this? One would expect that some genes would only be induced in the presence of a stressor, and it seems highly possible that the authors might be completely missing a relevant response by not looking at gene expression profiles in the presence of stress.

Response 5: As the reviewer pointed out, the expression of some genes would only be induced, or suppressed, in the presence of a stressor. In such cases, evolution under a stressor would alter the environment-dependent regulatory responses to achieve resistance. However, in this study, we neglected the environment-dependent expression changes and instead collected gene expression profiles only in the no-drug condition to compare expression profiles of the evolved strains without environmental-dependent biases. An alternative experimental design would be to quantify expression profiles of the evolved strains under the stress conditions to which the strains evolved in. However, in this strategy, the differences in expression profiles of the evolved strains can be obscured by the environment-specific regulatory responses. Note that the data obtained under the no-drug condition exhibited the expression profile clusters, as shown in Fig. 2, which may change or disappear in the presence of environment-specific biases. Ideally, it would be interesting to collect gene expression profiles under various environmental conditions, e.g., 192 strains \times 47 stress environments = 9024 conditions, to unveil both the environment-specific regulatory responses and their evolution. However, such analysis would be costly and outside the scope of this paper. To better explain the experimental design, we revised the Results and Discussion sections as follows:

Revision 5: Results (Page 6, Line 95-96)

“In the transcriptome analysis, all evolved strains were cultured without addition of stressors to standardize the culture condition.”

Revision 5: Discussion (Page 21-22, Line 45-460)

“A third limitation of our study was the limitation of our transcriptome analysis of the evolved strains to only non-stress conditions. Of course, the expression of some genes would only be induced or suppressed in the presence of a stressor. In such cases, evolution would change such environment-dependent regulatory responses to achieve resistance. However, in this study, we neglected the environment-dependent expression changes and collected gene expression profiles exclusively in the non-stress condition, to compare expression profiles of the evolved strains without environmental-dependent biases. An alternate choice of experimental design would be to collect the gene expression

profiles under various stress conditions, to reveal both the environment-specific regulatory responses and their evolution. However, such analysis is costly, and thus, will serve as our study's future scope."

Comment 6: There are several instances where the authors jump from high-level descriptions of the data to very specific examples, without explicitly communicating how representative the examples are. For instance, in Figure 3, why did the authors choose the pairs CBPC/AZT and TET/B.Cl.Ala vs other pairs? It would seem obvious that one can arbitrarily find highly- or lowly-correlated pairs.

Response 6: We thank the reviewer for pointing out these inconsistencies in our manuscript in which we introduced specific examples of the highly/lowly correlated stress pairs without explicit communication. We aimed to use examples of cross-resistance/collateral sensitivities within the stress pairs. We specifically introduced Fig. 3a to show that our analysis properly captured cross-resistance between stresses that have the same mechanism of action and Fig. 3b to show that certain negative correlations between stresses were observed in the evolved and mutant strains, indicating a trade-off. We further discuss this trade-off relationship, partially caused by the *rssB* mutation, later in the manuscript. To provide a more explicit explanation of this before introducing specific examples, we have modified the manuscript as follows:

Revision 6: Results (Page 11, Line 208-214)

"We recognized that some stress pairs showed high correlation in their resistance of the evolved strains. For example, evolved strains resistant to CBPC tended to exhibit resistance to aztreonam (AZT) as well ($R = 0.95$, Fig. 3a), both of which constitute β -lactam stresses. We then calculated correlation coefficients for the reconstructed mutant strains. Certain stress pairs, such as TET and B-Cl-Ala showed a negative correlation not only for the evolved strains, but also for the reconstructed mutant strains (Fig. 3b)."

Comment 7: For Figures 3D and S2a, why are values on the self-self diagonal zero (white)? One would expect those to be identically high (one).

Response 7: For Fig. 3d, the upper right of the plot shows the correlation coefficients among the mutant strains, while the lower left shows the correlation coefficients among the evolved strains. Therefore, we believe it is appropriate to leave the diagonal line blank since the two separated parts show correlation coefficients from a different population. However, we admit that Fig. 3d was confusing using the same color map for different population sets. We modified Fig.3d so that the correlation coefficients are shown in a different colormap to improve the clarity.

For Supplementary Fig. 2a, we left the self-self diagonal zero to highlight the observed cross resistances and collateral sensitivity relations among the evolved strains by showing the relative resistance for the pairs that were significant. However, we can indeed perform the same procedure

for the self-self diagonal elements as well. We thus updated Supplementary Fig. 2a showing the corresponding values for the self-self diagonal elements.

Fig. 3d, Pearson's correlation coefficient for all pairwise combinations of stress resistance for the evolved strains (upper right) and the site-directed mutants (lower left). The order of stresses was determined by hierarchical clustering performed on the pairwise correlation values of the site-directed mutants.

Supplementary Fig. 2a, Identified combinations of stresses that exhibited either cross resistance or collateral sensitivity for each of the four strains evolved in the same environment.

Comment 8: It is well appreciated that transport is a major resistance mechanism that induces non-specific protection from multiple stresses. Transport-related mutations appear over-represented in the authors' analyses and I wonder how much of the correlation depicted in Figure 3C is just due to transport-dependent resistance? In other words, do non-transport related mutations also have such strong correlations in phenotype between the evolved strains and constructed mutants?

Response 8: We appreciate the reviewer's interesting point. To address this comment, we have additionally plotted the relationship of the correlation coefficients for the evolved and mutant strains excluding the transport-related mutations: *dctA*, *uraA*, *sstT*, *livM*, *potA*, *oppA*, *cycA*, *yhjE*, *glpT*, *ompF*, *glnP*, *metN*, *ptsP*, *frlA*, *gabP*, *potH*, *mprA*, and *acrR* (Supplementary Fig. 7). The resulting correlation coefficient was $R = 0.55$, suggesting that non-transport related mutants also have strong correlations in phenotype with the evolved strains. Accordingly, we have added the following text to the revised manuscript.

Revision 8: Results (Page 12, Line 218-221)

"Interestingly, a high correlation was observed between the evolved strains and mutant strains excluding transporter related mutations (e.g. *acrR*, *ompF*), indicating that the high correlation in phenotype is not only caused by transport, which is a major mechanism of resistance (Supplementary Fig. 7)."

Supplementary Fig. 7. Phenotypic correspondence of the evolved strains and mutant strains without transporter related mutations

Relationships between the corresponding pairwise correlation coefficients for the evolved strains and the mutant strains. Here, transporter related mutations (*dctA*, *uraA*, *sstT*, *livM*, *potA*, *oppA*, *cycA*, *yhjE*, *glpT*, *ompF*, *glnP*, *metN*, *ptsP*, *frlA*, *gabP*, *potH*, *mprA*, and *acrR*) were excluded from the mutant strains.

Comment 9: The authors describe differences in acquired resistance as “decelerated evolution”. This seems imprecise, as the rate of resistance acquisition is expected to be set by the mutation rate and purifying selection, and the present work addresses neither of these. Instead, the authors are really just describing how different resistance alleles can confer different amounts of protection. Resistance alleles acquired by lab evolution are very sensitive to choice of selection methodology. Could this “deceleration” just be an artifact of their evolution scheme? In fact, ~300 total generations is a modest length for a lab evolution experiment.

Response 9: We appreciate the reviewer’s interesting comments. One possible explanation is that decreased evolution is caused by the difference in mutation frequencies between β -lactams and other stresses. To assess this possibility, we measured mutation frequencies under addition of IC_{50} concentrations of carbenicillin (CBPC), cefmetazole (CMZ) (both β -lactams), chloramphenicol (CP), tetracycline (TET), norfloxacin (NFLX), as well as in the absence of drugs. The results demonstrated that differences in mutation frequencies between β -lactams and other stresses were not observed (see figure below; Supplementary Fig. 10d in the revised manuscript), indicating that it is difficult to explain the observed decelerated evolution, at least for the evolved strains under CP, TET, and NFLX. At present, the molecular mechanism of this phenomenon remains unclear; however, we hypothesized that the inhibition of cell wall synthesis by β -lactams addition may disrupt membrane protein synthesis, including OmpF porin. Since the addition of β -lactams is known to induce bulge formation leading to cell lysis, a decrease in *ompF* expression and disruption of OmpF function by a mutation may not contribute to β -lactam resistance during such a deficient cell wall state. The phenomenon is clearly interesting and might contribute to a detailed understanding of β -lactam resistance evolution. Therefore, various possible mechanisms, including our hypothesis, should be investigated in the future.

Revision 9: Results (Page 18, Line 367-372)

“Since a link between mutation supply rate and adaptation rate was reported ⁴⁹, it is also possible that decreased evolution is caused by the difference in mutation frequencies between β -lactams and other stresses. To address this possibility, we measured mutation frequencies with β -lactams (CBPC and CMZ), or other drugs showing higher β -lactam resistance (CP, NFLX, TET); however, differences were not observed in mutation frequencies (Supplementary Fig. 10).”

Revision 9: Discussion (Page 22, Line 461-471)

“Finally, we identified decelerated evolution against β -lactams, however, examination of fitness cost, negative epistasis, and differences in mutation frequency were not sufficient to explain this phenomena. Currently, the associated molecular mechanism remains unclear; however, we hypothesize that the inhibition of cell wall synthesis by β -lactams addition may disrupt membrane protein synthesis, including OmpF porin. Since the addition of β -lactams reportedly induces bulge

formation leading to cell lysis⁵⁴, a decrease in *ompF* expression and disruption of OmpF function by a mutation, may not contribute to β -lactam resistance during such a deficient cell wall state. Nevertheless, this phenomenon is clearly interesting, and might contribute to a detailed understanding of β -lactam resistance evolution, however, the associated mechanisms, including our hypothesis, should be investigated as future works.”

Supplementary Fig. 10d

Mutation frequencies for MDS42 strains under addition of IC_{50} concentrations of β -lactam stresses (i.e. CBPC, CMZ) and other antibiotics in which the evolved strains acquired high resistance to β -lactam stresses (i.e. CP, NFLX, TET). Error bars represent the standard deviation of three independent experiments.

Comment 10: In their abstract, the authors state that they report a “novel constraint that leads to decelerated evolution”, but it is not at all clear what this novel constraint is. In fact, line 292 states “At present, the mechanism for the decelerated evolution against beta-lactams remains unclear.” This manuscript would overall benefit from a formal treatment on these “evolutionary constraints” as referenced in the title and abstract.

Response 10: We agree that the statement “novel constraint that leads to decelerated evolution” is not supported by our results as the “novel constraint” remains unclear. The abstract has been revised accordingly.

Revision 10: Abstract (Page 2, Line XX)

“We also report a decelerated evolution of β -lactam resistance, a phenomenon experienced by certain strains under various stresses resulting in higher acquired resistance to β -lactams compared to strains directly selected by β -lactams.”

Revision 10: Discussions (Page 22, Line 461)

“Finally, we identified decelerated evolution against β -lactams...”

Reply to Reviewer #3

Comment 0: Manuscript titled “High-throughput laboratory evolution reveals evolutionary constraints in *Escherichia coli*” describes experimental evolution of *Escherichia coli* MDS42 strain. Tomoya Maeda and co-workers evolved cells in 384-well plates, where each plate contained a modified M9 medium, glucose and an increasing concentration of a particular antimicrobial compound. They used automatic experimental system, developed previously by the Chikara Furusawa’s group, to evolve populations in 95 different antimicrobial compounds with 6 replicates, which means that they run in parallel a striking number of 576 plates. They experimentally evolved populations for 27 days and in evolved strains determined the transcriptome, whole-genome sequence and the half-maximal inhibitory concentration (IC50) of the antimicrobial compound that the strain was incubating with. Then they tested if specific mutations (identified with DNA sequencing) were responsible for the increased IC50 (relative to the parent), a phenotype that was measured in majority of evolved strains. Using an elegant method developed by Csaba Pal’s group they introduced these specific mutations into ancestor (parental) strain. They measured in many mutated parental strains a similar IC50 shift as measured in the evolved strain, thus demonstrating that specific mutations could be responsible for the increased IC50.

Studying genotype-phenotype mapping is an important research area, especially when a studied phenotype is a major health concern, which antimicrobial resistance (AMR) is. The high-throughput experimental approach that the group has developed is spectacular and the list of tested compounds is very well thought. I find evidence that specific mutations are shifting IC50 quite compelling.

However, certain aspects of the experimental design require an explanation. There is also a worrying lack of details regarding why they excluded so much data. I also believe that text and figures should be clearer and the entire manuscript should be better balanced with the existing knowledge on AMR. I hope my specific comments can improve the manuscript, because this solid and important work must come out and will be of interest to many AMR researchers.

Response 0: We would like to thank the reviewer for the constructive suggestions to improve our manuscript. We have revised the manuscript according to these suggestions. Our responses to specific comments are shown below.

1. Experimental design:

Comment 1.1: I wonder why you decided to use *E. coli* MDS42 strain. You probably wanted to constrain evolution to single-point mutations (thus avoid insertion sequences), and you perhaps wanted to work with evolved strains that experienced lower number of mutational events (given that MDS42 is promoted as anti-mutator strain). Whatever your motivation you need to explain it, and you have to discuss how this unusual strain might impact the generalisation of your results.

Response 1.1: The reason for our choice of the MDS42 strain in this study was to allow for easy and reliable genome resequencing analysis. Based on our past experiences, determining the precise position of IS transposition in *E. coli* is often difficult when we use short-read sequencers such as Illumina. This ambiguity in mutation identification can hinder the analysis of phenotype-genotype mapping. We, therefore, used MDS42 in this study, which allowed us to identify the precise relationship between resistance acquisition and mutation fixation, as shown in Fig. 2. Although certain essential factors in resistance evolution, including the effects of transposition and horizontal gene transfer, are difficult to analyze in our experimental setup, the precise phenotype-genotype mapping we identified will contribute to a better understanding of resistance evolution. Meanwhile, analysis of resistance evolution with transposition insertions and horizontal gene transfers will be a focus of our future works, likely based on long-read sequencing analysis such as PacBio and Oxford Nanopore. We have also included the following description in the revised Methods (Bacterial strains and growth media).

Revision 1.1: Methods (Page 22, Line 474-482)

“The insertion sequence (IS)-free *E. coli* strain MDS42⁵⁵ was purchased from Scarab Genomics (Scarab Genomics, Madison, Wisconsin, USA) and utilized throughout this study. The use of the IS elements-free strain facilitates reliable resequencing analysis results since the determination of the precise position of IS element insertions is often difficult using short-read sequencers. Although certain essential factors in resistance evolution, including the effects of transposition and horizontal gene transfer, are difficult to analyze in this experimental setup, use of this strain enables us to identify the precise correspondence between resistance acquisition and mutation fixation.”

Comment 1.2: In the past I did experiments with MDS42 strain, which I have not yet published. I can say that MDS42 has a poor growth in Davis minimal medium (DM) with glucose as the only source of carbon and energy (recipe for DM can be found in Lenski et al. 1991, American Naturalist). I have not tried to grow MDS42 in M9, but I wonder, is poor growth in a standard M9 medium the reason why you used a modified M9? I believe you should explain why you modified the medium.

Response 1.2: We chose modified M9 medium to facilitate simple media preparation (to avoid precipitation of metal ions). As described in line 503-506 in the main text, before laboratory evolution, the MDS42 strain was cultivated in the modified M9 medium without stressors for 96 h. This step enabled the MDS42 strain to adapt to this medium. We, therefore, believe our MDS42 strain can grow well on glucose minimal medium. The specific growth rate of our MDS42 strain after pre-cultivation (the parent strain for the laboratory evolution) in the modified M9 medium was 0.25 (1/h) and the MDS42 strain reached stationary phase at approximately 15 h cultivation. Although this growth rate was not considered fast, this parent strain grows sufficiently well for our laboratory evolution setup.

Revision 1.2: Methods (Page 24, Line 507-508)

“The specific growth rate of the parent strain in the modified M9 medium was 0.25 (1/h), which is sufficient for the laboratory evolution setup.”

Comment 1.3: To the best of knowledge, I cannot think of why you added sub-lethal concentration of erythromycin (15µm/ml) into modified M9 (page 18/line 367). I believe erythromycin is not effective against fungi or viruses (which could potentially contaminate your experiment). If you really used it, then you have to say why and discuss how is this relevant for the interpretation of your results. At the moment you are not discussing presence of erythromycin at all.

Response 1.3: Our preliminary laboratory evolution using the automated culture system failed due to contamination of *Ralstonia* sp likely derived from the liquid used in the automated culture system. This bacterium showed higher sensitivity to erythromycin than the *E. coli* MDS42 strain, therefore, we added 15 µg/mL erythromycin (approximately 1/10-fold concentration of IC₅₀ of *E. coli* MDS42) to the medium to avoid its contamination. We have also included the following description in the revised Methods (Bacterial strains and growth media).

Revision 1.3: Methods (Page 23, Line 485-488)

“To avoid contamination by other bacteria species, 15 ug/mL erythromycin (approximately 1/10-fold concentration of IC₅₀ of *E. coli* MDS42) was added to the medium throughout the experiments.”

Comment 1.4: You grew cells in 5 g of glucose per L (page 18/line 365). I expect that MDS42 will never use all that glucose in 24 hours, especially not on 34°C. In such conditions cells incubated with different antimicrobial compounds could grow differently not only in numbers but also in size. Your main measurement is OD, which is sensitive to cell's size. How much do you think is your IC₅₀ measurement sensitive to cell's size? Is it possible that at least some variation in observed IC₅₀ shifts could be down to variation in cell's size in the parent versus evolved strain?

Response 1.4: As the reviewer pointed out, our growth measurement by OD value depends on cell size and since the addition of antibiotics often changes bacterial cell shapes, the evolved strains may have altered their shape, which would have affected the IC₅₀ measurement. To evaluate this possibility, we examined the cell shapes of all 192 evolved strains when cultivated without the stressors. The cell size of the parent and evolved strains, estimated by microscope images, are shown below. Results show that some evolved strains changed cell shape (e.g., BZ evolved strains, and MMC evolved strains) resulting in either rounded or a shorter rod cell shape. Meanwhile, in the absence of stressors, no evolved strain showed filamentous/elongated cell shape with enlarged cell size. These results suggest that changes in cell size did not significantly affect our IC₅₀ measurement.

Revision 1.4: Methods (Page 33, Line 721-726)

“Note, this IC₅₀ measurement using OD₆₂₀ can be affected by changes in cell size among the evolved strains. To investigate this possibility, we quantified the cell size of all 192 evolved strains cultivated without the stressor and found that no evolved strains showed significantly elongated cell shape in the absence of stressors. We present the measured cell sizes for all 192 evolved strains and the parent strain in Supplementary Fig. 11 (Methods).”

Revision 1.4: Methods (Page 34, Line 740-745)

“**Quantifying single cell sizes of the evolved strains.** To evaluate the cell sizes of the parent and evolved strains, we acquired single cell images through an inverted microscope BX53 (Olympus, Tokyo, Japan). Single cell sizes were quantified using a custom ImageJ based code utilizing the “Threshold” and “Analyze Particles” functions. The mean and standard deviation of the cell size of approximately 160 cells for each evolved strain and the parent strain are provided in Supplementary Fig. 11.”

Supplementary Fig. 11. Area sizes of all 192 evolved and parent strains.

Mean area sizes for single cell observations of the evolved strains and parent strain. Error bars represent standard deviation.

2. Data exclusion:

Comment 2.1: I wonder why and how you selected 192 evolved strains (page 5/line 49). For instance,

I know for sure that you can experimentally evolve susceptible *E. coli* cells in non-lethal concentrations of trimethoprim (which is one of the environment you excluded).

Response 2.1: Among the 95 stress environments under which we performed laboratory evolution, we selected only 47 environments due to the limitation of experimental capacity. Three criteria were considered for this selection First, we selected environments to which a significant increase in IC₅₀ values was observed after evolution. Second, we selected a variety of stressors with different action mechanisms. Stresses with similar or identical action mechanisms were excluded, to analyze a wide variety of resistance acquisition mechanisms. Finally, we considered the novelty of expected results. For example, we excluded some antibiotics for which resistance acquisition mechanisms were already well studied. For instance, trimethoprim was excluded since the resistance evolution to it was analyzed in our previous study (Suzuki *et al.*, *Nature Comm.* **5** (1), 1-12 (2014)). We have included the following text to describe this process as follows:

Revision 2.1: Results (Page 5, Line 57-62)

“These 47 stressors were selected from the initial 95 stress environments, due to the limitation of experimental capacity. Selections were made based on the degree of increased IC₅₀ values; to ensure a variety of stressors with different action mechanisms; and based on the predicted novelty of the expected results.”

Comment 2.2: Also, why you analysed only 4 independent cultures from the same environment, because you started with six. In the text you only mention six cultures in the first result paragraph and on a few places in the supplementary material. I am convinced that you had good reasons to do it, but you have to explain them properly.

Response 2.2: We selected four independent cultures to analyze the effects of as many stresses on evolutionary dynamics as possible. The selected four culture lines were the top four showing higher IC₅₀ values among the six lines.

Revision 2.2: Results (Page 5, Line 61-62)

“For further analysis, we selected the top four independent culture lines showing higher IC₅₀ values among the six.”

Comment 2.3: On page 5 you mention GAH, NQO and MMC environments, where you detected higher number of mutations. Why you decided not to follow up this potentially important environments?

Response 2.3: We agree with your suggestion and are planning to follow up with such mutagens for future research. However, in this study, we would like to focus on understanding the constraints that shape the evolution of antibacterial chemicals resistance. We have, therefore, not discussed these environments further in the current study.

3. Clarity

Comment 3.1: It looks to me that 300 generations (page 5/line 45) is more like a rough estimate over the entire experiment. I think you have to explain how you come up with this number. As I understand, you transferred 5 μ l of diluted 24-hours culture into 45 μ l of a fresh medium, where the 5 μ l inoculum were prepared by diluting an overnight culture to OD₆₂₀ of 0.0003. Surely ODs of overnight cultures were not the same across environments, which means that the number of generations per day could vary not only per environment but also across the experimental evolution. Usually, if you dilute cultures 100-fold, the number of generations per day is 6.64.

Response 3.1: First, we apologize for this error in the Materials and Methods. In our laboratory evolution, the initial OD₆₂₀ value was 0.00015, although we described it as 0.0003 in the original manuscript. Second, in our daily propagation, the cells were transferred from a well at which OD₆₂₀ exceeds a threshold (= 0.09), while when the culture becomes saturated, the OD₆₂₀ value reaches approximately 0.2. From the initial and final OD values, the number of generations per 24 hours is estimated between 9.2 and 10.4. Thus, the 27-day propagation corresponds to approximately 249–280 generations. To explain the estimated number of generations, we revised the manuscript as follows.

Revision 3.1: Results (Page 5, Line 49-51)

“In total, 576 independent culture series were maintained (95 stressors plus a control without any stressor \times six replicates) for 27 daily passages corresponding to approximately 250–280 generations.”

Revision 3.1: Methods (Page 24, Line 513-516)

“The OD₆₂₀ values of the precultures were measured using the automated culture system, and precultured cells, calculated to have initial OD₆₂₀ values of 0.00015, were inoculated into each well (5 μ L of diluted overnight culture into 45 μ L of medium per well) of the 384-well microplates and cultivated with agitation at 300 rotations/min at 34 °C.”

Revision 3.1: Methods (Page 24, Line 518-521)

“The automated culture system selected the defined well with the highest chemical concentration in which cells could grow; the cells in the selected well were diluted to an OD₆₂₀ of 0.00015 and transferred to new plates containing fresh medium and chemical gradients.”

Comment 3.2: There is some inconsistency when you reported the number of control replicates that were compared to evolved strains. You started with six independent parental lines (page 4/line 44), but then you say on page 5 (line 50) "...plus a control without any stress". Then on page 26 (lines 564 & 568) you stated that "...the mean of 13 independent replicas of the parent strain".

Response 3.2: We apologize for these inconsistent descriptions. We began laboratory evolution with six independent lines. We subsequently selected four of the six evolved strains for further analysis. To clarify the difference in IC₅₀ values between the parent and evolved strain, only for the parent strain, we quantified IC₅₀ values for 13 independent cultures, not four cultures, as in the case of the evolved strains. To explain the number of samples adequately, we revised the manuscript as follows.

Revision 3.2: Results (Page 5, Line 61-62)

"For further analysis, we selected the top four independent culture lines showing higher IC₅₀ values among the six."

Revision 3.2: Methods (Page XX, Line 563-564)

"The relative IC₅₀ values were computed by comparing the IC₅₀ of each evolved strain to the mean of 13 independent measurements of the MDS42 parent strain."

Comment 3.3: Sentence that starts on page 5/line 62 should be moved to a more appropriate place, probably where transcriptome data is first introduced.

Response 3.3:

We thank the reviewer for this suggestion. We moved the applicable sentences to the section that first introduces the transcriptome analysis as follows:

Revision 3.3: Results (Page 6, Line 93-96)

"Phenotypic changes of the evolved strains were quantified by transcriptome analysis to examine gene expression levels responsible for stress resistance (Supplementary Table 4). In the transcriptome analysis, all evolved strains were cultured without addition of stressors to standardize the culture condition."

Comment 3.4: On page 5 (line 67) you need to be more specific. When you say "about 80%", is this 80% out of 47 or 95 environments. You need to report the number of strains with some measure of variance. And what is the variation in the number of mutations among strains evolved in the same environment?

Response 3.4: We thank the reviewer for pointing out this unclear description. Among the 192 sequenced strains, 147 (76.6%) harbored fewer than five mutations. To show the number distribution of mutations, we added Supplementary Fig. 3 to the revised manuscript (shown below), in which the mean and standard deviation of mutation number in the same environments are presented. To better describe this, we revised the manuscript as follows.

Revision 3.4: Results (Page 6, Line 77-80)

“Although some of these strains carried more than 20 mutations, 147/192 evolved strains (76.6%) harbored fewer than five. To show the variation in the number of mutations among strains evolved in the same environment, the mean number and standard deviation of mutations for the four strains are shown in Supplementary Fig. 3.”

Supplementary Fig. 3. Number of mutations identified in the evolved strains. The mean number of identified mutations for the four evolved strains in each environment are shown. The error bars represent standard deviation.

Comment 3.5: On page 5 (line 70) you mentioned only strains evolved in GAH, but what about NQO and MMC. It would be better if you tell us the mean with CI for all three environments.

Response 3.5: Among the 47 stressors we inspected, the highest number of mutations was observed in glutamic acid γ -hydrazide (GAH) evolved strains carrying 157 ± 67 (standard deviation) mutations. These numbers were much larger than that of the evolved strains against known mutagens (e.g. 4-nitroquinoline-1-oxide (NQO, 23 ± 5 mutations) and mitomycin C (MMC, 27 ± 4 mutations)), indicating a high mutagenic activity of GAH. Following the reviewer’s suggestion, we revised the manuscript as follows.

Revision 3.5: Results (Page 6, Line 80-85)

“Among the 47 stressors, the highest number of mutations was observed in glutamic acid γ -hydrazide (GAH) evolved strains carrying 157 ± 67 mutations (Supplementary Fig. 3), which was significantly more than the number observed in evolved strains against known mutagens (e.g. 4-nitroquinoline-1-oxide (NQO, 23 ± 5 mutations) and mitomycin C (MMC, 27 ± 4 mutations)), indicating a high mutagenic activity of GAH.”

Comment 3.6: Figure 1c (page 6/line 74) is slightly misleading. First column probably shows 21 “synonymous” mutations (it is hard to be sure given the small size of the panel). So, probably, “other point mutations”, “duplications”, “indels” and “large deletions” (next four columns) are all non-synonymous mutations? Perhaps you could show only four columns (four types) and have two colors (synonymous & non-synonymous).

Response 3.6: As the reviewer pointed out, our original classification of mutations was confusing, and did not explicitly note non-synonymous mutations. We have, therefore, updated Fig. 1c so that the point mutations are classified into synonymous mutations, non-synonymous mutations (mutation that result in a single amino acid change in the coding region), and other point mutations. This additional classification also increased the visibility of the plot by decreasing the y-range. We thank the reviewer for their kind suggestion.

Fig. 1c, Distribution of mutation events for the evolved strains according to mutation type, except for strains evolved in GAH, NQO, and MMC. Other point mutations include those in intergenic/noncoding regions.

Comment 3.7: I do not understand where is 80% coming from (page 6/line 76). Please explain the number.

Response 3.7: We have estimated the ratio of beneficial mutations using the ratio of nonsynonymous to synonymous mutations per site (dN/dS) which was 5.26 for the current experiment. This was performed according to the procedure outlined in the supplement of Tenaillon *et al. Science*

335(6067), 457-461, (2012). The ratio of beneficial mutations y can be calculated under the assumption that dN/dS should be 1.0 under strict neutrality. This led to our estimation of $y = (5.26 - 1.0)/5.26 = 0.810$. We have added this detailed procedure to the Methods section.

Revision 3.7: Methods (Page 32, Line 708-713)

“Estimating the ratio of beneficial mutations. To estimate the ratio of beneficial mutations of the evolved strains, we used the ratio of nonsynonymous to synonymous mutations per site (dN/dS) which was 5.26 for the current experiment. Here, we followed the procedure introduced in the supplement of Tenaillon et al., (2012) ²⁹. The ratio of beneficial mutations ‘ y ’ can be calculated under the assumption that dN/dS should be 1.0 under strict neutrality. This leads to our estimation of $y = (5.26 - 1.0)/5.26 = 0.810$.”

Comment 3.8: I do not think you have enough evidence to describe stressors that you included as non-mutagens (page 6/ line 74). As far as I know at least 6-mercaptopurine monohydrate, acriflavine, hydrogen peroxide, nitrofurantoin were shown to be mutagenic in bacteria (and this is probably not an exhaustive list).

Response 3.8: As the reviewer noted, our description “For strains evolved in non-mutagens” was inappropriate since previously known mutagen stressors, such as 6-MP, were included here as non-mutagens. Our aim here was to exclude strains that had an especially high number of mutations to increase the reliability of the estimate of the ratio of non-synonymous to synonymous mutations per site. We have thus modified the manuscript as follows:

Revision 3.8: Results (Page 6, Line 85-90)

“Excluding strains evolved in GAH, NQO, and MMC, which harbored more than 18 mutations on average per stressor, 21 and 307 mutations were identified as synonymous and nonsynonymous mutations, respectively (Fig. 1c). For strains evolved in stresses other than GAH, NQO and MMC, the ratio of nonsynonymous to synonymous mutations per site was 5.26, implying that approximately 80% of the nonsynonymous mutations were beneficial ^{29,30}.”

Comment 3.9: You need to properly define what you mean by commonly mutated gene (page 7/ line 114).

Response 3.9: We apologize for our original incomplete description and have since defined a commonly mutated gene as when any type of mutation in the same gene was found in at least two out of four independent culture lines evolved under the same environment. We have revised the text accordingly, as follows:

Revision 3.9: Results (Page 11, Line 198-200)

“Commonly mutated genes were defined as mutations in the same gene identified in a minimum of two of the four independent culture lines evolved under the same environment.”

Comment 3.10: On page 10 (line 173) you mentioned 64 representative mutations. Are these mutations located only in those 25 genes? Have you introduced mutations also from the rest of 188 mutated genes?

Response 3.10: We selected the 64 representative mutations from commonly mutated genes and/or candidate mutations conferring drug resistance in each environment. Thus, the 64 representative mutations include mutations that were located in genes other than the 25 genes as well. The mutations attempted to introduce were the 64 mutated genes, as well as five other mutations (*cysP*, *hpt*, *yfdZ*, *pstB* and *secB*). Both the reconstructed *cysP* and *hpt* mutant strains showed amino acid auxotrophy, therefore, we excluded these from further analysis. Mutations in *cysP* and *hpt* genes were commonly identified in the sodium dichromate (SDC) and 6-mercaptopurine (6-MP) evolved strains, respectively. The *cysP* mutant strain grown in the modified M9 medium supplemented with 20 amino acids did not show increased IC₅₀ value, while the *hpt* mutant strain grown in the modified M9 medium supplemented with 20 amino acids showed an 18-fold increase in 6-MP resistance (Supplementary Materials). Although the *yfdZ* mutation was commonly found in ABU evolved strains, the reconstructed *yfdZ* mutant strain did not exhibit resistance to ABU (see the supplementary text). BZ and EDTA evolved strains commonly had mutations in *pstB* and *secB*, respectively. Additionally, although we attempted to construct both *pstB* and *secB* disrupted mutants, we were unsuccessful.

Comment 3.11: On page 11 / line 196 you reported results from the erythromycin environment. It is important to say something about erythromycin being present in your medium.

Response 3.11: Related to your comment 1.3, we added erythromycin to avoid *Ralstonia* sp. contamination. Among the 192 evolved strains, except erythromycin evolved strains, NMNOE5, NMNOE6, NVAE5, ABUE6, and M9E3 strains had mutations in *ycbZ* which are commonly mutated genes in erythromycin environments and conferred 3.4-fold increased erythromycin resistance. The reconstructed *ycbZ* mutant strain also showed a 2.1-fold and 2.5-fold increase in NMNO and NVA resistance, respectively. Therefore, the fixation of *ycbZ* mutations in NMNO and NVA environments could result from its fitness gain to these stressors. In contrast, the *ycbZ* mutant strain did not show increased ABU resistance (Table S2). Meanwhile, one out of four control experiments, without any additional stressor (M9E3), carried the *ycbZ* mutation. Overall, *ycbZ* mutations were not commonly identified mutations, however, the presence of 1/10 IC₅₀ erythromycin may have partially affected the resistance evolution. We have included the following corresponding description to the revised manuscript:

Revision 3.11: Methods (Page 15, Line 305-311)

“Note, one out of four control experiments without any additional stressor (M9E3 strain) also carried the *ycbZ* mutation for EM. Additionally, ABUE6 evolved strains carried the *ycbZ* mutation although the *ycbZ* disruption did not confer ABU resistance (Table S2). These mutations in M9E3 and ABUE6 strains may have arisen from the application of 1/10 IC₅₀ erythromycin to all culture medium, to avoid contamination, which may have caused adaptive evolution to occur in response to the low concentrations of erythromycin.”

Comment 3.12: On page 15 / line 286 you reported very marginal p value that I would not necessarily trust. You also do not mention statistical test or number of observations. I believe presenting N, measure of variation and identity of the statistical test is a gold standard for each report.

Response 3.12: As the reviewer noted, the p value we report for the number of identified *ompF* mutations is marginal. We therefore, modified the manuscript as follows. We also apologize for not mentioning the statistical test nor the number of observations. We revised the manuscript, adding the specific statistical test (Fisher’s exact test) and the number of observations (N = 32).

Revision 3.12: Results (Page 17-18, Line 358-363)

“In contrast, the strains evolved under CBPC or CMZ had fewer mutations in *OmpF* related genes (one out of eight evolved strains) in comparison with other strains with high β -lactam resistance ($p = 0.04$, Fisher’s exact test, N = 32). This result might suggest that in our laboratory evolution setup, the fixation of mutations related to *OmpF* is suppressed under the addition of β -lactams, even though they can increase their resistance to the drug.”

Comment 3.13: Regarding the decelerated evolution, I think you should discuss the potential effect of mutation supply rate on your results. A link between mutation supply rate and adaptation rate as shown in this paper (Salverda et al. Proceedings of the National Academy of Sciences 114, 12773-12778, 2017), might be relevant.

Response 3.13: We appreciate the reviewer’s interesting comments One possible explanation is that decreased evolution is caused by the difference in mutation frequencies between β -lactams and other stresses. To address this possibility, we measured mutation frequencies under the addition of IC₅₀ concentrations of carbenicillin (CBPC), cefmetazole (CMZ) (both β -lactams), chloramphenicol (CP), tetracycline (TET), and norfloxacin (NFLX), as well as in the absence of drugs. The results demonstrated that mutation frequencies between β -lactams and other stresses do not differ (see figure below; Supplementary Fig. 10d in the revised manuscript). This indicated that the change in mutation frequency does not effectively explain the observed decelerated evolution, at least for the

evolved strains under CP, TET, and NFLX. At present, the molecular mechanism of this phenomenon remains unclear. We hypothesized that the inhibition of cell wall synthesis by β -lactam addition may disrupt membrane protein synthesis, including that of OmpF porin. Since the addition of β -lactams is known to induce bulge formation leading to cell lysis, a decrease in *ompF* expression and disruption of OmpF function via mutation, may not contribute to β -lactam resistance during such a deficient cell wall state. The phenomenon is clearly interesting and might contribute to a detailed understanding of β -lactam resistance evolution. Hence, the possible mechanisms, including our hypothesis, should be investigated further.

Revision 3.13: Results (Page 18, Line 367-372)

“Since a link between mutation supply rate and adaptation rate was reported ⁴⁹, it is also possible that decreased evolution is caused by the difference in mutation frequencies between β -lactams and other stresses. To address this possibility, we measured mutation frequencies with β -lactams (CBPC and CMZ), or other drugs showing higher β -lactam resistance (CP, NFLX, TET); however, differences were not observed in mutation frequencies (Supplementary Fig. 10).”

Revision 3.13: Discussion (Page 22, Line 461-471)

“Finally, we identified decelerated evolution against β -lactams, however, examination of fitness cost, negative epistasis, and differences in mutation frequency were not sufficient to explain this phenomena. Currently, the associated molecular mechanism remains unclear; however, we hypothesize that the inhibition of cell wall synthesis by β -lactams addition may disrupt membrane protein synthesis, including OmpF porin. Since the addition of β -lactams reportedly induces bulge formation leading to cell lysis ⁵⁴, a decrease in *ompF* expression and disruption of OmpF function by a mutation, may not contribute to β -lactam resistance during such a deficient cell wall state. Nevertheless, this phenomenon is clearly interesting, and might contribute to a detailed understanding of β -lactam resistance evolution, however, the associated mechanisms, including our hypothesis, should be investigated as future works.”

Supplementary Fig. 10d

Mutation frequencies for MDS42 strains under addition of IC₅₀ concentrations of β -lactam stresses (i.e. CBPC, CMZ) and other antibiotics in which the evolved strains acquired high resistance to β -lactam stresses (i.e. CP, NFLX, TET). Error bars represent the standard deviation of three independent experiments.

Comment 3.14: Please, you have to help the reader to better understand so many of your abbreviations. For instance, you mention B.Cl.Ala on page 15 / line 307 (but defined it on page 13 / line 241). Also names on page 17 (line 334) are defined way back in the Result section. I also suggest that you try to use as many standard abbreviations as possible, I believe 5-Fluoroorotic Acid is usually abbreviated as 5-FOA.

Response 3.14: As the reviewer pointed out, our paper includes many abbreviations that may prove cumbersome for the readers. We have, therefore, modified the manuscript as follows. We have also replaced the following abbreviations 3.AT, 5.FOA, 5.FU, 6.MP, B.Cl.Ala, to 3-AT, 5-FOA, 5-FU, 6-MP, B-Cl-Ala, respectively.

Revision 3.14: Discussion (Page 19, Line 388-390)

“For instance, we found that various antibiotics with different action mechanisms exhibited collateral sensitivity to metabolic inhibitors including l-valine (LVAL), β -chloro-l-alanine (B-Cl-Ala), and glutamic acid γ -hydrazide (GAH).”

Comment 3.15: Statement on page 16 / line 324 is probably an overstatement, because you have cases where mutation introduction was not enough (which you acknowledged later on page 17 / line 348), so I would rephrase it.

Response 3.15: As the reviewer noted, there were certain features of the evolved strains that could not be fully explained by the single mutant strains. We have, therefore, edited the manuscript as follows.

Revision 3.15: Discussion (Page 19-20, Line 405-410)

“The pairwise correlation coefficients between stresses, indicating cross-resistance and collateral sensitivity, observed in the reconstructed mutants agreed with those of the evolved strains (Fig. 3c, d). Although correlation coefficients are only capable of probing the averaged directionality evolution, these results suggest that the observed evolutionary constraints for resistance in the evolved strains were rooted in acquired mutations.”

Comment 3.16: On page 19 / line 404 you mentioned that the IC₅₀ of the isolated clone was almost identical. This is important information, so give this statement a proper weight by reporting both means with CI.

Response 3.16: We thank the reviewer for this suggestion. Accordingly, we revised the manuscript as follows. We have also added the IC₅₀ data for the endpoint population in supplemental Table 2.

Revision 3.16: Methods (Page 24-25, Line 524-526)

“We have further confirmed that the IC₅₀ of the isolated clone was nearly identical to that of the corresponding population in the endpoint culture, where the mean of IC₅₀ (isolated clone) - IC₅₀ (endpoint culture) was 0.18 ± 0.22 (95% confidence interval).”

Comment 3.17: Page 20 / line 413: Did you mean OD₆₀₀ or OD₆₂₀.

Response 3.17: For laboratory evolution and IC₅₀ measurements, we measured OD₆₂₀ by using FilterMax F5 microplate reader (Molecular Devices) which is connected to the automated culture system. When we collected cells for total RNA extraction, we used a different microplate reader (ARVO microplate reader, PerkinElmer Inc.) measuring OD₆₀₀. Hence, the description is correct.

4. More than just terminology

Comment 4.1: I think it is important that you introduce distinctions between heteroresistance, persistence, tolerance and genetic resistance (as defined in Balaban et al. 2019, Nature Reviews Microbiology). On short, genetic resistance is an increase in minimal inhibitory concentration (MIC) due to a resistance mutation or a plasmid-borne resistance gene, whereas heteroresistance entails a MIC increase without obtaining genetic resistance (e.g. higher expression of efflux pumps). Tolerance is an increase in the minimum duration to kill 99% of the population (MDK99) and persisters are a subpopulation of dormant tolerant cells with a high MDK99. Also, resistant and heteroresistant cells divide in the presence of the antibiotic, while tolerant cells and persisters do not. I strongly suggest that you consistently use the terminology from this seminal paper, written by AMR pioneers aiming to decrease confusion in AMR research.

Response 4.1: We appreciate the reviewer's valuable comments. Accordingly, we revised the main text to introduce distinctions between heteroresistance and genetic resistance. While we agree that persistence and tolerance are also very important mechanisms for survival under drug exposure, we did not discuss them in the present study since analyses pertaining to duration of drug exposure and survival behavior of the subpopulation are difficult to incorporate in our experimental setup. We revised the main text carefully to use the terminology from the papers written by AMR pioneers. For more detail, see the response to comment 4.2 below.

Comment 4.2: In your study it is not clear what is actually being affected by mutations, but I would say that across all environments cells mostly evolved heteroresistance. For instance, you can enrich the bacterial culture with tolerant cells by pre-exposing them to antibiotics that arrest protein synthesis (Kwan et al. *Antimicrob. Agents Chemother.* 2013), like rifampicin (RFP), tetracycline (TET) and carbonyl cyanide 3-chlorophenylhydrazone (CCCP), all three compounds were also part of your study. Just to make a point, for the rifampicin I am convinced that you have not sequenced the resistant strain, because genetic resistance to rifampicin is possible only via ~79 point mutations within an *rpoB* gene (Garibyan et al., *DNA Repair* 2, 593-608). What you probably evolved in rifampicin environment is a heteroresistant strain with the 1.5-fold increased MIC. Note that Kwan's paper is published in 2013, when tolerance was still confused with heteroresistance.

Response 4.2: We appreciate the reviewer's valuable comments. Accordingly, we conducted a population analysis profile (PAP) to examine whether the commonly identified mutations confer genetic resistance or heteroresistance. We defined heteroresistance, according to previous studies (El-Halfawy & Valvano, *Clinical Microbiol. Rev.*, **28**, 191-207, (2015); Nicoloff et al., *Nat. Microbiol.*, **4**, 504-514 (2019)), as strains with resistant subpopulations growing at frequencies of 1×10^{-7} or higher in antibiotic concentrations at least 8-fold higher than the highest non-inhibitory concentration of the wild type. Although we could not examine all possible combinations, we conducted a PAP of 33 combinations of drug \times the reconstructed mutant strain pairs (Supplementary Fig. 8). These combinations were selected according to the mutation and drug pairs showing more than eightfold increased IC_{50} value, the representative mutations in the supervised PCA classes (*acrR* in class 5, *gyrA* in class 8, *mprA* in class 1, *ompF* in class 2 and 10, *prfF* (renamed from *sohA*) in class 11, and *rssB* in class 9), and the novel resistance-conferring mutations which we identified (*gshA*, *ycbZ* and *yhjE*). The results suggest that heteroresistance is a common resistance phenotype among our laboratory evolved strains, as shown in previous studies using clinical isolates (Band et al., *Nat. Microbiol.*, **4**, 1627-1635 (2019); Nicoloff et al., *Nat. Microbiol.*, **4**, 504-514 (2019)). According to these new results, we revised the main text as follows.

Revision 4.2: Introduction (Page 4, Line 32-37)

"To examine whether the whole population or only a subset of cells were phenotypically resistant, we conducted a population analysis profile (PAP). Heteroresistance is a common phenomenon for several bacterial species, and antibiotic classes, in which a subpopulation among susceptible cells exhibits increased resistance²⁴⁻²⁷. Here, we identified many heteroresistance-conferring mutations in novel genes, as well as known repressors for multidrug efflux pumps."

Revision 4.2: Results (Page 12-13, Line 224-244)

“To examine whether the commonly identified mutations confer genetic resistance or heteroresistance, we conducted a PAP of 33 drug combinations and the reconstructed mutant strain pairs (Supplementary Fig. 8). These combinations were selected according to the mutation and drug pairs exhibiting > 8-fold increase in IC₅₀ value, the representative mutations in the supervised PCA classes (*acrR* in class 5, *gyrA* in class 8, *mprA* in class 1, *ompF* in class 2 and 10, *prfF* in class 11, and *rssB* in class 9), and the novel resistance-conferring mutations that we had identified (*gshA*, *ycbZ* and *yhjE*). Heteroresistance strains are defined as strains with resistant subpopulations growing at frequencies of 1×10^{-7} or higher in antibiotic concentrations at least 8-fold higher than the highest non-inhibitory concentration of the wild type^{27,39}. Among the 33 combinations, 11 were categorized as genetic resistance including seven cross-genetic resistances; while 15 combinations were categorized as heteroresistance, including 12 cross-heteroresistance pairs (Supplementary Fig. 8). Although, seven combinations exhibited a 2-fold increase compared to the highest non-inhibitory concentration, there were no resistant subpopulations at antibiotic concentrations 8-fold higher. Regarding the supervised PCA classes, the *ompF* mutant showed cross-genetic resistance, while the *acrR*, *mprA*, *prfF*, and *rssB* mutant strains exhibited cross-heteroresistances (Supplementary Fig. 8). Alternatively, the *gyrA* mutant strains showed both cross-genetic resistance and cross-heteroresistance (Supplementary Fig. 8). These results suggest that heteroresistance is a common resistance phenotype among our laboratory evolved strains, which agrees with results of previous studies using clinical isolates^{26,27}.”

Revision 4.2: Results (Page 13-14, Line 260-265)

“Since the reconstructed *acrR*, and the *mprA* mutant strains, exhibited cross-heteroresistance to EM and TET, or GAH and TET respectively (Supplementary Fig. 8), the enhanced drug efflux caused by *acrR* or *mprA* mutations can cause cross-heteroresistance. Similarly, it was previously reported that overexpression of the efflux pump results in heteroresistance in several pathogenic bacteria²⁴.”

Revision 4.2: Results (Page 14, Line 270-272)

“Since the reconstructed *ompF* mutant strain showed cross-genetic resistance to CBPC, FTD, and NVA (Supplementary Fig. 8), inactivation of the OmpF porin can cause cross-genetic resistance.”

Revision 4.2: Results (Page 14, Line 278-280)

“Moreover, the DVAL and NVA resistances by *yhjE* inactivation constituted heteroresistance (Supplementary Fig. 8). Taken together, our results indicate that the effects of chemical uptake and efflux are major mechanisms for cross-genetic resistance and heteroresistance.”

Revision 4.2: Results (Page 288-290, Line 15)

“Furthermore, the *prfF* mutant strain showed cross-heteroresistances to 5-FOA and CBPC indicating that the *prfF* mutation conferred heteroresistance (Supplementary Fig. 8).”

Revision 4.2: Results (Page 15, Line 304-305)

“Moreover, the *ycbZ* mutant strain showed cross-heteroresistances to EDTA and NVA, indicating that the *ycbZ* mutation confers heteroresistance (Supplementary Fig. 8).”

Supplementary Fig. 8. Both genetic resistant and heteroresistant strains were observed within the mutant strains.

Colony-forming units of the parent strain and mutant strains for increasing concentrations of a respective drug. The CFU for each strain is normalized by the CFU of the no drug condition. Here, the highest non-inhibitory concentration, represented by the concentration where the CFU of the parent strain is 50% less than that of the no drug condition, is set to 1.

Comment 4.3: My suggestion is that you rewrite your results and discussion by really focusing on novel connections that you found. And whenever you confirmed an already known connection, you

should cite a relevant paper. Regarding the tolerance/heteroresistance/resistance dilemma that I just raised, I would double check again all the genes in all the environments and make sure that the right term is used for describing the evolutionary outcome.

Response 4.3: Related to your comment C35 and C41, we rewrote our results and discussion to focus on our new findings. See the revised manuscript and the response to your comments C35 and C41.

Comment 4.4: I do not agree with your definition of cross-resistance and collateral sensitivity on page 3 (line 10), because double genetic resistance is a very rare event. I believe acquisition of resistance (and heteroresistance) to a certain drug can be accompanied by higher or lower heteroresistance (or tolerance) to another drug.

Response 4.4: Related to your comment C41, we revised our definition of cross-resistance and collateral sensitivity as you suggested. Please refer to our response to your comment C41. We also inserted the following description to clarify our revised definition.

Revision 4.4: Results (Page 5, Line 69-70)

“Here, cross-resistance refers to both cross-genetic resistance and cross-heteroresistance.”

Comment 4.5: At the end of the discussion you listed other explanations named as “multiple mutations”, “epistasis” and “non-genetic adaptation”. I am not convinced that they are really an alternative, but whatever your idea is you need to be more specific and give us some references. For instance, epigenetics in AMR has been proposed although it is not in a mainstream (Ghosh et al. *Antimicrob. Agents Chemother.* 64, e02225-02219). Also there is a lot of literature linking epistasis with AMR (see please Wong, A. *Front. Microbiol.* 8, 246-246).

Response 4.5: We appreciate the reviewer’s critical comments. Since the evolved strains had several mutations that are potentially involved in resistance acquisition, we believe that multiple mutations and epistasis are possible explanations. According to Reviewer’s comment, we revised this section of the manuscript and cited the appropriate references.

Revision 4.5: Discussion (Page 21, Line 437-447)

“The differences between the evolved strains and reconstructed mutant strains might suggest the contribution of multiple mutations, and epistasis among them, to the resistance changes since the evolved strains had several mutations that are potentially involved in resistance acquisition. Moreover, the ability of certain resistance-conferring mutations to impose a different degree of drug resistance on different genetic backgrounds⁵², might reflect the ubiquity of epistatic interactions among genetic

alterations. There may have also been a contribution made by non-genetic adaptation, which is difficult to explain by the phenotype-genotype mapping presented in this study. Meanwhile, epigenetic changes e.g. methylation of bacterial DNA, can influence gene expression and/or mutation rates resulting in resistance acquisition⁵³.”

Reviewers' Comments:

Reviewer #1:

Remarks to the Author:

The authors have done a good job to answer the raised concerns. They performed additional analyses or adequately altered or toned down the claims. My concerns on the choice of stress conditions used and their clinical relevance remain, but this is an issue that goes beyond the scope of the current manuscript.

Reviewer #2:

Remarks to the Author:

In this submission, Maeda et al have made several revisions which overall improve the manuscript and address many of my original concerns. I find most of the revisions addressing my technical concerns satisfactory.

That said, I continue to have concerns on conceptual pieces of the study design and exposition. For instance, in response to my original Comment #6, in which I ask why the authors chose to focus on the specific examples on their (now) lines 208-311, the authors do not clarify for me the logical basis for why these specific examples were highlighted for analysis, instead of others; and what the overall message from these analyses are. Clearly the authors will identify novel cross- and hetero-resistance phenotypes, because they are performing a larger lab evolution experiment than has been previously published, but it is lost on me what the biological significance of any of these findings are, in light of the extensive literature on cross- and hetero-resistance.

I have similar concerns in relation to my original Comment #9 dealing with "decelerated beta-lactam evolution". It seems that this analysis is intended to be purely observational, as the authors end the section by stating that the mechanisms for this are unknown. It is therefore again difficult to ascertain what is actually learned from these extensive analyses, without any specific mechanistic follow-up to explain why the resulting phenotypes are observed.

Simply put- this is an impressive body of work, and the data will be of broad interest to the AMR lab evolution research community, but in its current form this manuscript is only descriptive with no specific validated predictions and I am unsure what I learn from these impressive efforts.

This is all related to my original Comment #10, in which I am unsure that the authors have met the criteria they set forth in their abstract. This work is motivated by a "systematic investigation of evolutionary constraints" but I am hard-pressed to understand how this work systematically investigates evolutionary constraints such as positive/negative selection, mutation rate, culture conditions, etc. The abstract states "identification of novel trade-off relationships associated with drug resistance" and I am unsure what is novel about the observation that cross- and hetero-resistance can be found in large-scale lab evolution experiments, in the absence of mechanistic experiments.

Reviewer #3:

Remarks to the Author:

Dear authors, I find your rebuttal very clear and well organised, thank you.

You did well in dealing with my comments and I have nothing more to add.

I believe performing all that additional experiments in these turbulent times is a great achievement, so well done.

I look forward to reading your follow-up papers.

Good luck!

Rok Krašovec

Reply to Reviewer #1

Comment 0: The authors have done a good job to answer the raised concerns. They performed additional analyses or adequately altered or toned down the claims. My concerns on the choice of stress conditions used and their clinical relevance remain, but this is an issue that goes beyond the scope of the current manuscript.

Response 0: We thank the reviewer for his/her valuable comments to improve our manuscript. As the reviewer points out, we seek to investigate the clinical relevance of laboratory evolved strains in future works.

Reply to Reviewer #2

Comment 0: In this submission, Maeda et al have made several revisions which overall improve the manuscript and address many of my original concerns. I find most of the revisions addressing my technical concerns satisfactory.

Response 0: We would like to thank the reviewer again for his/her valuable comments to improve our manuscript.

Comment 1: That said, I continue to have concerns on conceptual pieces of the study design and exposition. For instance, in response to my original Comment #6, in which I ask why the authors chose to focus on the specific examples on their (now) lines 208-311, the authors do not clarify for me the logical basis for why these specific examples were highlighted for analysis, instead of others; and what the overall message from these analyses are. Clearly the authors will identify novel cross- and hetero-resistance phenotypes, because they are performing a larger lab evolution experiment than has been previously published, but it is lost on me what the biological significance of any of these findings are, in light of the extensive literature on cross- and hetero-resistance.

Response 1:

As the reviewer points out, in lines 208-311, we presented several specific examples of cross- and hetero-resistance phenotypes and their responsible genes, which we identified by our high-throughput laboratory evolution and mutant reconstructions. We chose these examples since they were new findings and thus were worth discussing in the main text. It should be emphasized here that we have also provided a comprehensive description of the identified mutations and resistance mechanisms for all evolved strains (Supplementary Discussion) and the effects of 64 mutations on resistance/sensitivity to 47 stresses (Supplementary Data 5). Although these results also included new findings and provide clues for future studies, we could not present them in the main text due to the length limitation. We believe that sharing the results of our large-scale analysis will contribute to future studies in the field of antibiotic resistance evolution. To explain the above points, we revised the manuscript as follows.

Revision 1: Discussion (Page 20) lines 421-427

Although we only highlighted a limited number of resistance mechanisms, a comprehensive description of the mutations identified in the stresses used in this study is given in the Supplementary Discussion. We believe that sharing our results in this manuscript, including identified mutations, transcriptome changes, and resistance profiles in the evolved strains, as well as phenotypic changes in the reconstructed mutants, will provide clues for future studies and contribute to the field of antibiotic resistance evolution.

Comment 2: I have similar concerns in relation to my original Comment #9 dealing with "decelerated beta-lactam evolution". It seems that this analysis is intended to be purely observational, as the authors end the section by stating that the mechanisms for this are unknown. It is therefore again difficult to ascertain what is actually learned from these extensive analyses, without any specific mechanistic follow-up to explain why the resulting phenotypes are observed.

Response 2:

As the reviewer pointed out, the mechanism for "decelerated beta-lactam evolution" remains unclear and the presentation of the result is observational. However, this result is a novel and interesting phenomenon, and thus we have judged that this observation is worth sharing. It should also be noted that we have tried to identify the mechanisms for the decelerated evolution by analyzing the fitness cost by *ompF* mutation, epistatic interactions between *ompF* and another mutation, and alternation of mutation frequency by adding beta-lactams, as shown in Supplementary Figure 10. Although we found that these hypotheses are difficult to explain the observed decelerated evolution, we expect that sharing the negative results will contribute to future studies to clarify this interesting phenomenon. To explain the above points, we revised the manuscript as follows.

Revision 2: Discussion (Page 22-23) lines 472-483

Finally, we identified the decelerated evolution against β -lactams. At present, the mechanism for the observed decelerated evolution remains unclear. We tested several hypotheses to explain this phenomenon, including the effect of fitness cost, negative epistasis, and alternation of mutation frequency by adding β -lactams, and found that they were not sufficient to explain it. However, we further hypothesize that the inhibition of cell wall synthesis by β -lactams addition may disrupt membrane protein synthesis, including OmpF porin. Since the addition of β -lactams reportedly induces bulge formation leading to cell lysis, a decrease in *ompF* expression and disruption of OmpF function by a mutation, may not contribute to β -lactam resistance during such a deficient cell wall state. Nevertheless, this phenomenon is clearly interesting, and we expect our observation and testing of several hypotheses will contribute to future studies to unveil the dynamics of antibiotic resistance evolution.

Comment 3: Simply put- this is an impressive body of work, and the data will be of broad interest to the AMR lab evolution research community, but in its current form this manuscript is only descriptive with no specific validated predictions and I am unsure what I learn from these impressive efforts.

Response 3:

The present study is not designed to validate existing hypotheses. Instead, we tried to provide new findings by the large-scale analysis. It should be emphasized that our high-throughput laboratory evolution had yielded various findings on antibiotic resistance evolution, including cross- and hetero-resistance and genetic bases for them. Although this paper might be seen as descriptive because of presenting many results, various hypotheses on resistance mechanisms generated by phenotype and genotype analyses of evolved strains were validated by mutant reconstructions. We believe that sharing our results in this manuscript, including identified mutations, transcriptome changes, and resistance profiles in the evolved strains, as well as phenotypic changes in the reconstructed mutants, will provide clues for future studies and contribute to the field of antibiotic resistance evolution. To explain the above points, we revised the manuscript as follows.

Revision 3: Discussion (Page 20) lines 421-427

Although we only highlighted a limited number of resistance mechanisms, a comprehensive description of the mutations identified in the stresses used in this study is given in the Supplementary Discussion. We believe that sharing our results in this manuscript, including identified mutations, transcriptome changes, and resistance profiles in the evolved strains, as well as phenotypic changes in the reconstructed mutants, will provide clues for future studies and contribute to the field of antibiotic resistance evolution.

Comment 4: This is all related to my original Comment #10, in which I am unsure that the authors have met the criteria they set forth in their abstract. This work is motivated by a "systematic investigation of evolutionary constraints" but I am hard-pressed to understand how this work systematically investigates evolutionary constraints such as positive/negative selection, mutation rate, culture conditions, etc. The abstract states "identification of novel trade-off relationships associated with drug resistance" and I am unsure what is novel about the observation that cross- and hetero-resistance can be found in large-scale lab evolution experiments, in the absence of mechanistic experiments.

Response 4:

We agree that the evolutionary constraints can be affected by several conditions, including the strength of selection pressure, mutation rate, culture conditions, and so forth. Therefore, we need to investigate such conditions for a complete understanding of evolutionary constraints. Nevertheless, even though we used a unified condition for selection, mutation rate, and culture condition, our high-throughput laboratory evolution under various stress environments contributed to unveil the evolutionary constraints of *E. coli*. We identified resistance-conferring mutations, cross-resistance/collateral sensitivity relationships (e.g., *rpoS* associated metabolic inhibitor sensitivity, and *prfF* associated oxidative stress sensitivity). These results by this large-scale study

can be a basis for further analysis of how the evolutionary dynamics of *E. coli* is constrained. The problem of how these factors are affected by other conditions will be an important topic and remain as future works in the field of antibiotic resistance evolution. Considering the reviewer's comment, we added the following description to the Discussion.

Revision 4: Discussion (Page 20) lines 427-430

Of course, these findings can be affected by various conditions, including the strength of selection pressure, mutation frequency, and culture condition. The problem of how the identified genotypic and phenotypic alterations are affected by other conditions will be an important topic and remain as future works.

Reply to Reviewer #3

Comment 0:

Dear authors, I find your rebuttal very clear and well organised, thank you.

You did well in dealing with my comments and I have nothing more to add.

I believe performing all that additional experiments in these turbulent times is a great achievement, so well done.

I look forward to reading your follow-up papers.

Good luck!

Rok Krašovec

Response 0: We would like to thank the reviewer again for his valuable comments to improve our manuscript.